# Cold Diffusion: Inverting Arbitrary Image Transforms Without Noise

**Arpit Bansal**[1]    **Eitan Borgnia**[*1]    **Hong-Min Chu**[*1]    **Jie S. Li**[1]
**Hamid Kazemi**[1]    **Furong Huang**[1]    **Micah Goldblum**[2]
**Jonas Geiping**[1]    **Tom Goldstein**[1]
[1]University of Maryland    [2]New York University

## Abstract

Standard diffusion models involve an image transform – adding Gaussian noise – and an image restoration operator that inverts this degradation. We observe that the generative behavior of diffusion models is not strongly dependent on the choice of image degradation, and in fact an entire family of generative models can be constructed by varying this choice. Even when using completely deterministic degradations (e.g., blur, masking, and more), the training and test-time update rules that underlie diffusion models can be easily generalized to create generative models. The success of these fully deterministic models calls into question the community's understanding of diffusion models, which relies on noise in either gradient Langevin dynamics or variational inference, and paves the way for generalized diffusion models that invert arbitrary processes.

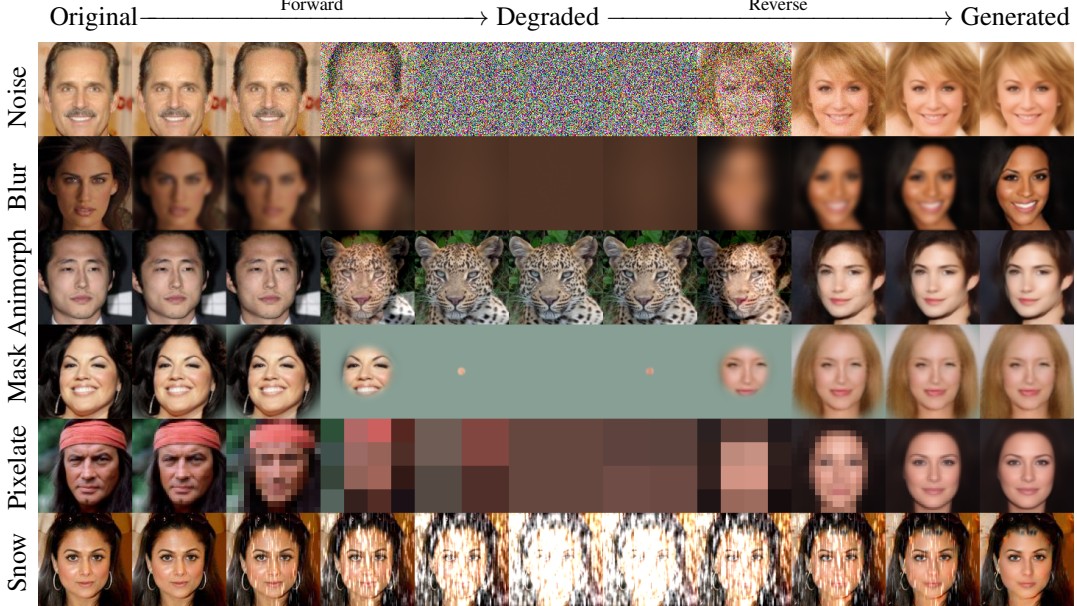

Figure 1: Demonstration of the forward and backward processes for both hot and cold diffusions. While standard diffusions are built on Gaussian noise (top row), we show that generative models can be built on arbitrary and even noiseless/cold image transforms, including the ImageNet-C *snowification* operator, and an *animorphosis* operator that adds a random animal image from AFHQ.

37th Conference on Neural Information Processing Systems (NeurIPS 2023).

# 1  Introduction

Diffusion models have recently emerged as powerful tools for generative modeling [Ramesh et al., 2022]. Diffusion models come in many flavors, but all are built around the concept of random noise removal; one trains an image restoration/denoising network that accepts an image contaminated with Gaussian noise, and outputs a denoised image. At test time, the denoising network is used to convert pure Gaussian noise into a photo-realistic image using an update rule that alternates between applying the denoiser and adding Gaussian noise. When the right sequence of updates is applied, complex generative behavior is observed.

The origins of diffusion models, and also our theoretical understanding of these models, are strongly based on the role played by Gaussian noise during training and generation. Diffusion has been understood as a random walk around the image density function using *Langevin dynamics* [Sohl-Dickstein et al., 2015, Song and Ermon, 2019], which requires Gaussian noise in each step. The walk begins in a high temperature (heavy noise) state, and slowly anneals into a "cold" state with little if any noise. Another line of work derives the loss for the denoising network using variational inference with a Gaussian prior [Ho et al., 2020, Song et al., 2021a, Nichol and Dhariwal, 2021].

In this work, we examine the need for Gaussian noise, or any randomness at all, for diffusion models to work in practice. We consider *generalized diffusion models* that live outside the confines of the theoretical frameworks from which diffusion models arose. Rather than limit ourselves to models built around Gaussian noise, we consider models built around arbitrary image transformations like blurring, downsampling, etc. We train a restoration network to invert these deformations using a simple $\ell_p$ loss. When we apply a sequence of updates at test time that alternate between the image restoration model and the image degradation operation, generative behavior emerges, and we obtain photo-realistic images.

The existence of *cold diffusions* that require no Gaussian noise (or any randomness) during training or testing raises questions about the limits of our theoretical understanding of diffusion models. It also unlocks the door for potentially new types of generative models with very different properties than conventional diffusion seen so far.

# 2  Background

Both the Langevin dynamics and variational inference interpretations of diffusion models rely on properties of the Gaussian noise used in the training and sampling pipelines. From the score-matching generative networks perspective [Song and Ermon, 2019, Song et al., 2021b], noise in the training process is critically thought to expand the support of the low-dimensional training distribution to a set of full measure in ambient space. The noise is also thought to act as data augmentation to improve score predictions in low density regions, allowing for mode mixing in the stochastic gradient Langevin dynamics (SGLD) sampling. The gradient signal in low-density regions can be further improved during sampling by injecting large magnitudes of noise in the early steps of SGLD and gradually reducing this noise in later stages.

Kingma et al. [2021] propose a method to learn a noise schedule that leads to faster optimization. Using a classic statistical result, Kadkhodaie and Simoncelli [2021] show the connection between removing additive Gaussian noise and the gradient of the log of the noisy signal density in deterministic linear inverse problems. Here, we shed light on the role of noise in diffusion models through theoretical and empirical results in applications to inverse problems and image generation.

Iterative neural models have been used for various inverse problems [Romano et al., 2016, Metzler et al., 2017]. Recently, diffusion models have been applied to them [Song et al., 2021b] for the problems of deblurring, denoising, super-resolution, and compressive sensing [Whang et al., 2021, Kawar et al., 2021, Saharia et al., 2021, Kadkhodaie and Simoncelli, 2021].

Although not their focus, previous works on diffusion models have included experiments with deterministic image generation [Song et al., 2021a, Dhariwal and Nichol, 2021, Karras et al., 2022] and in selected inverse problems [Kawar et al., 2022]. Recently, Rissanen et al. [2022] use a combination of Gaussian noise and blurring as a forward process for diffusion. Though they show the feasibility of a different degradation, here we show definitively that noise is not a *necessity* in diffusion models, and we observe the effects of removing noise for a number of inverse problems.

Despite prolific work on generative models in recent years, methods to probe the properties of learned distributions and measure how closely they approximate the real training data are by no means closed fields of investigation.

Indirect feature space similarity metrics such as Inception Score [Salimans et al., 2016], Mode Score [Che et al., 2016], Frechet inception distance (FID) [Heusel et al., 2017], and Kernel inception distance (KID) [Bińkowski et al., 2018] have been proposed and adopted to some extent, but they have notable limitations [Barratt and Sharma, 2018]. To adopt a popular frame of reference, we will use FID as the feature similarity metric for our experiments.

# 3 Generalized Diffusion

Standard diffusion models are built around two components. First, there is an image degradation operator that contaminates images with Gaussian noise. Second, a trained restoration operator is created to perform denoising. The image generation process alternates between the application of these two operators. In this work, we consider the construction of generalized diffusions built around arbitrary degradation operations. These degradations can be randomized (as in the case of standard diffusion) or deterministic.

## 3.1 Model components and training

Given an image $x_0 \in \mathbb{R}^N$, consider the *degradation* of $x_0$ by operator $D$ with severity $t$, denoted $x_t = D(x_0, t)$. The output distribution $D(x_0, t)$ of the degradation should vary continuously in $t$, and the operator should satisfy $D(x_0, 0) = x_0$.

In the standard diffusion framework, $D$ adds Gaussian noise with variance proportional to $t$. In our generalized formulation, we choose $D$ to perform various other transformations such as blurring, masking out pixels, downsampling, and more, with severity that depends on $t$. We explore a range of choices for $D$ in Section 4.

We also require a *restoration* operator $R$ that (approximately) inverts $D$. This operator has the property that $R(x_t, t) \approx x_0$. In practice, this operator is implemented via a neural network parameterized by $\theta$. The restoration network is trained via the minimization problem

$$\min_\theta \mathbb{E}_{x \sim \mathcal{X}} \|R_\theta(D(x, t), t) - x\|,$$

where $x$ denotes a random image sampled from distribution $\mathcal{X}$ and $\|\cdot\|$ denotes a norm, which we take to be $\ell_1$ in our experiments. We have so far used the subscript $R_\theta$ to emphasize the dependence of $R$ on $\theta$ during training, but we will omit this symbol for simplicity in the discussion below.

## 3.2 Sampling from the model

After choosing a degradation $D$ and training a model $R$ to perform the restoration, these operators can be used in tandem to invert severe degradations by using standard methods borrowed from the diffusion literature. For small degradations ($t \approx 0$), a single application of $R$ can be used to obtain a restored image in one shot. However, because $R$ is typically trained using a simple convex loss, it yields blurry results when used with large $t$. Rather, diffusion models [Song et al., 2021a, Ho et al., 2020] perform generation by iteratively applying the denoising operator and then adding noise back to the image, with the level of added noise decreasing over time. This is the standard update sequence in Algorithm 1.

---
**Algorithm 1** Naive Sampling (Eg. DDIM)

> **Input:** A degraded sample $x_t$
> **for** $s = t, t - 1, \ldots, 1$ **do**
>     $\hat{x}_0 \leftarrow R(x_s, s)$
>     $x_{s-1} = D(\hat{x}_0, s - 1)$
> **end for**
> **Return:** $x_0$

---
**Algorithm 2** Transformation Agnostic Cold Sampling (TACoS)

> **Input:** A degraded sample $x_t$
> **for** $s = t, t - 1, \ldots, 1$ **do**
>     $\hat{x}_0 \leftarrow R(x_s, s)$
>     $x_{s-1} = x_s - D(\hat{x}_0, s) + D(\hat{x}_0, s - 1)$
> **end for**

---

When the restoration operator is perfect, *i.e.* when $R(D(x_0, t), t) = x_0$ for all $t$, one can easily see that Algorithm 1 produces exact iterates of the form $x_s = D(x_0, s)$. But what happens for imperfect

restoration operators? In this case, errors can cause the iterates $x_s$ to wander away from $D(x_0, s)$, and inaccurate reconstruction may occur.

We find that the standard sampling approach in Algorithm 1 (explained further in A.8) works well for noise-based diffusion, possibly because the restoration operator $R$ has been trained to correct (random Gaussian) errors in its inputs. However, we find that it yields poor results in the case of cold diffusions with smooth/differentiable degradations as demonstrated for a deblurring model in Figure 2. We propose Transformation Agnostic Cold Sampling (TACoS) in Algorithm 2, which we find to be superior for inverting smooth, cold degradations.

This sampler has important mathematical properties that enable it to recover high quality results. Specifically, for a class of linear degradation operations, it can be shown to produce exact reconstruction (*i.e.* $x_s = D(x_0, s)$) even when the restoration operator $R$ fails to perfectly invert $D$. We discuss this in the following section.

### 3.3 Properties of TACoS

It is clear from inspection that both Algorithms 1 and 2 perfectly reconstruct the iterate $x_s = D(x_0, s)$ for all $s < t$ if the restoration operator is a perfect inverse for the degradation operator. Hence in this section, we will discuss the reconstruction operator that fails to reconstruct the image perfectly i.e. incurs error. We first analyze the stability of these algorithms to errors in the restoration operator and then theoretically show that for a simple blur degradation, the error incurred using algorithm 1 is always greater than algorithm 2.

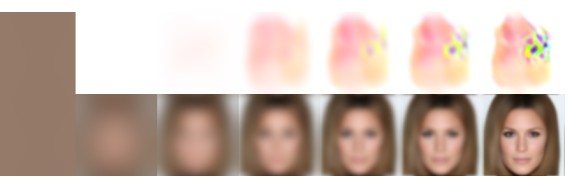

Figure 2: Comparison of sampling methods for unconditional generation using cold diffusion on the CelebA dataset. Iterations 2, 4, 8, 16, 32, 64, 128, 192, and 256 are presented. **Top:** Algorithm 1 produces compounding artifacts and fails to generate a new image. **Bottom:** TACoS succeeds in sampling a high quality image without noise.

For small values of $x$ and $s$, TACoS as described in 2 is tolerant of error in the restoration operator $R$. To see why, consider a problem with linear degradation function of the form $D(x, s) \approx x + s \cdot e$ for a constant vector $e$. We choose this ansatz because the Taylor expansion of any smooth degradation $D(x, s)$ around $x = x_0, s = 0$ has the form $D(x, s) \approx x + s \cdot e(x) + \text{HOT}$ where HOT denotes higher order terms. Note, however, the analysis below requires $e$ to be a constant that does not depend on $x$. The constant/zeroth-order term in this Taylor expansion is zero because we assumed above that the degradation operator satisfies $D(x, 0) = x$.

For a degradation $D(x, s)$ and any restoration operator $R$, the term $x_{s-1}$ in TACoS becomes

$$x_s - D(R(x_s, s), s) + D(R(x_s, s), s - 1) = D(x_0, s) - D(R(x_s, s), s) + D(R(x_s, s), s - 1)$$
$$= x_0 + s \cdot e - R(x_s, s) - s \cdot e + R(x_s, s) + (s - 1) \cdot e = x_0 + (s - 1) \cdot e = D(x_0, s - 1)$$

By induction, we see that the algorithm outputs the value $x_s = D(x_0, s)$ for all $s < t$, regardless of the choice of $R$. In other words, for *any* choice of $R$, the iteration behaves the same as it would when $R$ is a perfect inverse for the degradation $D$.

By contrast, Algorithm 1 does not enjoy this behavior even for small values of $s$. In fact, when $R$ is not a perfect inverse for $D$, $x_0$ is not a fixed point of the update rule in Algorithm 1 because $x_0 \neq D(R(x, 0), 0) = R(x, 0)$ and hence errors compound. If $R$ does not perfectly invert $D$, we should expect Algorithm 1 to incur errors, even for small values of $s$. Meanwhile, for small values of $s$, the behavior of $D$ approaches its first-order Taylor expansion, and Algorithm 2 becomes immune to errors in $R$. Figure 2 demonstrates the stability of TACoS described in Algorithm 2 vs. Algorithm 1 for a deblurring model.

The above analysis is not a complete convergence theory but rather highlights a desirable theoretical property of our method that a naive sampler lacks. However, we can prove that for a *toy* problem in which the blur operator removes one frequency at a time, the error incurred by sampling using Algorithm 1 is greater than the error incurred from using Algorithm 2. We present the proof of this claim in A.9.

# 4 Generalized Diffusions with Various Transformations

In this section, we take the first step towards cold diffusion by reversing different degradations and hence performing conditional generation. We will extend our methods to perform unconditional (i.e. from scratch) generation in Section 5. We emprically evaluate generalized diffusion models trained on different degradations with TACoS proposed in Algorithm 2. We perform experiments on the vision tasks of deblurring, inpainting, and super-resolution. We perform our experiments on MNIST [LeCun et al., 1998], CIFAR-10 [Krizhevsky, 2009], and CelebA [Liu et al., 2015]. In each of these tasks, we gradually remove the information from the clean image, creating a sequence of images such that $D(x_0, t)$ retains less information than $D(x_0, t-1)$. For these different tasks, we present both qualitative and quantitative results on a held-out testing dataset and demonstrate the importance of the sampling technique described in Algorithm 2. For all quantitative results in this section, the Frechet inception distance (FID) scores [Heusel et al., 2017] for degraded and reconstructed images are measured with respect to the testing data. Additional information about the quantitative results, convergence criteria, hyperparameters, and architecture of the models presented below can be found in the appendix.

## 4.1 Deblurring

We consider a generalized diffusion based on a Gaussian blur operation (as opposed to Gaussian noise) in which an image at step $t$ has more blur than at $t-1$. The forward process given the Gaussian kernels $\{G_s\}$ and the image $x_{t-1}$ at step $t-1$ can thus be written as

$$x_t = M_t \circ x_{t-1} = M_t \circ \ldots \circ M_1 \circ x_0 = \bar{M}_t \circ x_0 = D(x_0, t)$$

where $*$ denotes the convolution operator, which blurs an image using a kernel.

We train a deblurring model by minimizing the loss (1), and then use TACoS to invert this blurred diffusion process for which we trained a DNN to predict the clean image $\hat{x}_0$. Qualitative results are shown in Figure 3 and quantitative results in Table 1. Qualitatively, we can see that images created using the sampling process are sharper and in some cases completely different as compared to the direct reconstruction of the clean image. Quantitatively we can see that the reconstruction metrics such as RMSE and PSNR get worse when we use the sampling process, but on the other hand FID with respect to held-out test data improves. The qualitative improvements and decrease in FID show the benefits of the generalized sampling routine, which brings the learned distribution closer to the true data manifold. Note: we compare the images reconstructed via Algorithm 2, with direct generation as compared to Algorithm 1. This is because the image reconstruction via Algorithm 1 is much worse than both direct generation and Algorithm 2. Nevertheless, to back our claim, we present their results in A.10.

In the case of blur operator, the sampling routine can be thought of adding frequencies at each step. This is because the sampling routine involves the term $D(\hat{x}_0, t) - D(\hat{x}_0, t-1)$ which in the case of blur becomes $\bar{G}_t * x_0 - \bar{G}_{t-1} * x_0$. This results in a difference of Gaussians, which is a band pass filter and contains frequencies that were removed at step $t$. Thus, in the sampling process, we sequentially add the frequencies that were removed during the degradation process.

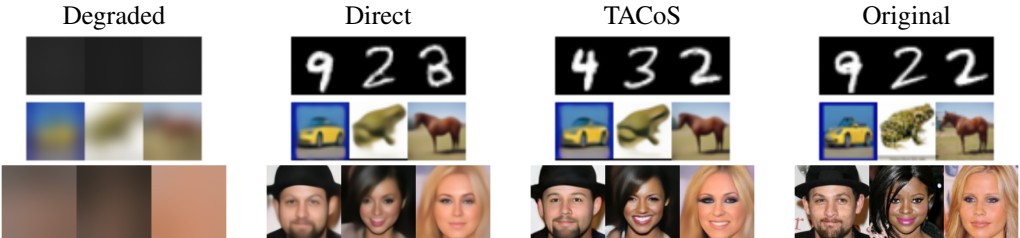

Figure 3: Deblurring models trained on the MNIST, CIFAR-10, and CelebA datasets. **Left to right:** degraded inputs $D(x_0, T)$, direct reconstruction $R(D(x_0, T))$, sampled reconstruction with TACoS described in Algorithm 2, and original image.

Table 1: Quantitative metrics for quality of image reconstruction using deblurring models.

| Dataset | Degraded | | | Sampled | | | Direct | | |
|---|---|---|---|---|---|---|---|---|---|
| | FID | SSIM | RMSE | FID | SSIM | RMSE | FID | SSIM | RMSE |
| MNIST | 438.59 | 0.287 | 0.287 | **4.69** | 0.718 | 0.154 | 5.10 | **0.757** | 0.142 |
| CIFAR-10 | 298.60 | 0.315 | 0.136 | **80.08** | 0.773 | 0.075 | 83.69 | **0.775** | 0.071 |
| CelebA | 382.81 | 0.254 | 0.193 | **26.14** | 0.568 | 0.093 | 36.37 | **0.607** | 0.083 |

## 4.2 Inpainting

We define a schedule of transforms that progressively grays-out pixels from the input image. We remove pixels using a Gaussian mask as follows: For input images of size $n \times n$ we start with a 2D Gaussian curve of variance $\beta$, discretized into an $n \times n$ array. We normalize so the peak of the curve has value 1, and subtract the result from 1 so the center of the mask as value 0. We randomize the location of the Gaussian mask for MNIST and CIFAR-10, but keep it centered for CelebA. We denote the final mask by $z_\beta$.

Input images $x_0$ are iteratively masked for $T$ steps via multiplication with a sequence of masks $\{z_{\beta_i}\}$ with increasing $\beta_i$. We can control the amount of information removed at each step by tuning the $\beta_i$ parameter. In the language of Section 3, $D(x_0, t) = x_0 \cdot \prod_{i=1}^{t} z_{\beta_i}$, where the operator $\cdot$ denotes entry-wise multiplication.

Figure 4 presents results on test images and compares the output of the inpainting model to the original image. The reconstructed images display reconstructed features qualitatively consistent with the context provided by the unperturbed regions of the image. We quantitatively assess the effectiveness of the inpainting models on each of the datasets by comparing distributional similarity metrics before and after the reconstruction. Our results are summarized in Table 2. Note, the FID scores here are computed with respect to the held-out validation set.

| Degraded | Direct | TACoS | Original |
|---|---|---|---|

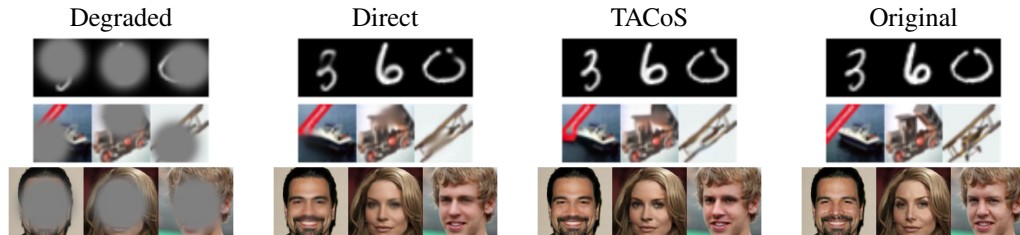

Figure 4: Inpainting models trained on the MNIST, CIFAR-10, and CelebA datasets. **Left to right:** Degraded inputs $D(x_0, T)$ , direct reconstruction $R(D(x_0, T))$, sampled reconstruction with TACoS described in Algorithm 2, and original image.

Table 2: Quantitative metrics for quality of image reconstruction using inpainting models.

| Dataset | Degraded | | | Sampled | | | Direct | | |
|---|---|---|---|---|---|---|---|---|---|
| | FID | SSIM | RMSE | FID | SSIM | RMSE | FID | SSIM | RMSE |
| MNIST | 108.48 | 0.490 | 0.262 | **1.61** | 0.941 | 0.068 | 2.24 | **0.948** | 0.060 |
| CIFAR-10 | 40.83 | 0.615 | 0.143 | **8.92** | 0.859 | 0.068 | 9.97 | **0.869** | 0.063 |
| CelebA | 127.85 | 0.663 | 0.155 | **5.73** | 0.917 | 0.043 | 7.74 | **0.922** | 0.039 |

## 4.3 Super-Resolution

For this task, the degradation operator downsamples the image by a factor of two in each direction. The final resolution of $x_T$ is 4×4 for MNIST and CIFAR-10 and 2×2 in the case of Celeb-A. After each down-sampling, the lower-resolution image is resized to the original image size, using nearest-neighbor interpolation. More details are available in Appendix A.3

Figure 5 presents example testing data inputs for all datasets and compares the output of the super-resolution model to the original image. Though the reconstructed images are not perfect for the

more challenging datasets, the reconstructed features are qualitatively consistent with the context provided by the low resolution image. Table 3 compares the distributional similarity metrics between degraded/reconstructed images and test samples.

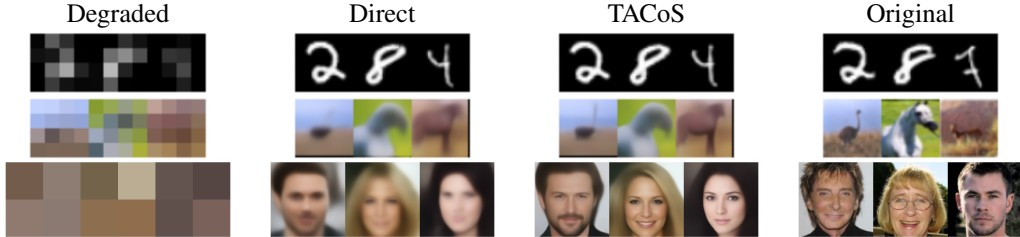

| Degraded | Direct | TACoS | Original |
|---|---|---|---|

Figure 5: Superresolution models trained on the MNIST, CIFAR-10, and CelebA datasets. **Left to right:** degraded inputs $D(x_0, T)$, direct reconstruction $R(D(x_0, T))$, sampled reconstruction with TACoS described in Algorithm 2, and original image.

Table 3: Quantitative metrics for quality of image reconstruction using super-resolution models.

| Dataset | Degraded | | | Sampled | | | Direct | | |
|---|---|---|---|---|---|---|---|---|---|
| | FID | SSIM | RMSE | FID | SSIM | RMSE | FID | SSIM | RMSE |
| MNIST | 368.56 | 0.178 | 0.231 | 4.33 | 0.820 | 0.115 | **4.05** | **0.823** | 0.114 |
| CIFAR-10 | 358.99 | 0.279 | 0.146 | **152.76** | 0.411 | 0.155 | 169.94 | **0.420** | 0.152 |
| CelebA | 349.85 | 0.335 | 0.225 | **96.92** | 0.381 | 0.201 | 112.84 | **0.400** | 0.196 |

# 5 Cold Generation

Diffusion models can successfully learn the underlying distribution of training data, and thus generate diverse, high quality images [Song et al., 2021a, Dhariwal and Nichol, 2021, Jolicoeur-Martineau et al., 2021, Ho et al., 2022]. We will first discuss deterministic generation using Gaussian noise and then discuss in detail unconditional generation using deblurring. Finally, we provide a proof of concept that the TACoS described in Algorithm 2 can be extended to other degradations.

## 5.1 Generation using deterministic noise degradation

Here we discuss image generation using a noise-based degradation presented in our notation from Section 3, which we will later prove is equivalent to DDIM [Song et al., 2021a]. We use the following degradation operator: $D(x, t) = \sqrt{\alpha_t}x + \sqrt{1 - \alpha_t}z$.

$D$ is an interpolation between the data point $x$ and a sampled noise pattern $z \in \mathcal{N}(0, 1)$. During training, $D$ is applied *once* and thus $z$ is sampled once for every image in every batch. However, sampling involves iterative applications of the degradation operator $D$, which poses the question of how to pick $z$ for the sequence of degradations $D$ applied in a single image generation.

There are three possible choices for $z$. The first would be to resample $z$ for each application of $D$, but this would make the sampling process nondeterministic for a fixed starting point. Another option is to sample a noise pattern $z$ *once* for each *separate* image generation and reuse it in each application of $D$. In Table 4 we refer to this approach as *Fixed Noise*. Finally, one can calculate the noise vector $z$ to be used in step $t$ of reconstruction by using the formula

$$\hat{z}(x_t, t) = \frac{x_t - \sqrt{\alpha_t}R(x_t, t)}{\sqrt{1 - \alpha_t}}.$$

This method denoted *Estimated Noise* in Table 4 turns out to be equivalent to the deterministic sampling proposed in Song et al. [2021a]. We discuss this equivalence in detail in Appendix A.6.

## 5.2 Image generation using blur

The forward diffusion process in noise-based diffusion models has the advantage that the degraded image distribution at the final step $T$ is simply an isotropic Gaussian. One can therefore perform

(unconditional) generation by first drawing a sample from the isotropic Gaussian, and sequentially denoising it with backward diffusion.

When using blur as a degradation, the fully degraded images do not form a nice closed-form distribution that we can sample from. They do, however, form a simple enough distribution that can be modeled with simple methods. Note that every image $x_0$ degenerates to an $x_T$ that is constant (i.e., every pixel is the same color) for large $T$. Furthermore, the constant value is exactly the channel-wise mean of the RGB image $x_0$, and can be represented with a 3-vector. This 3-dimensional distribution is easily represented using a Gaussian mixture model (GMM). This GMM can be sampled to produce the random pixel values of a severely blurred image, which can be deblurred using cold diffusion to create a new image.

Our generative model uses a blurring schedule where we progressively blur each image with a Gaussian kernel of size $27 \times 27$ over 300 steps. The standard deviation of the kernel starts at 1 and increases exponentially at the rate of 0.01. We then fit a simple GMM with one component to the distribution of channel-wise means. To generate an image from scratch, we sample the channel-wise mean from the GMM, expand the 3D vector into a $128 \times 128$ image with three channels, and then apply TACoS.

Empirically, the presented pipeline generates images with high fidelity but low diversity, as reflected quantitatively by comparing the perfect symmetry column with results from hot diffusion in Table 4. We attribute this to the perfect correlation between pixels of $x_T$ sampled from the channel-wise mean Gaussian mixture model. To break the symmetry between pixels, we add a small amount of Gaussian noise (of standard deviation 0.002) to each sampled $x_T$. As shown in Table 4, the simple trick drastically improves the quality of generated images. We also present the qualitative results for cold diffusion using blur transformation in Figure 6, and further discuss the necessity of TACoS proposed in Algorithm 2 for generation in Appendix A.7.

Table 4: FID scores for CelebA and AFHQ datasets using hot (noise) and cold diffusion (blur transformation). Breaking the symmetry within pixels of the same channel further improves FID.

| Dataset | Hot Diffusion | | Cold Diffusion | |
| --- | --- | --- | --- | --- |
| | Fixed Noise | Estimated Noise | Perfect symmetry | Broken symmetry |
| CelebA | 59.91 | 23.11 | 97.00 | 49.45 |
| AFHQ | 25.62 | 20.59 | 93.05 | 54.68 |

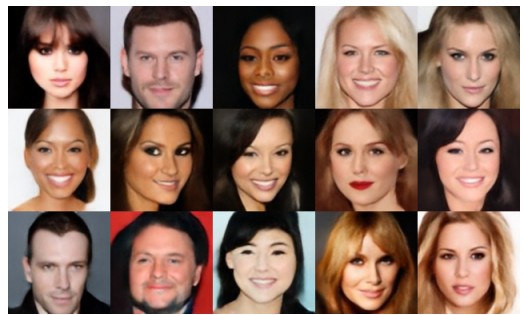 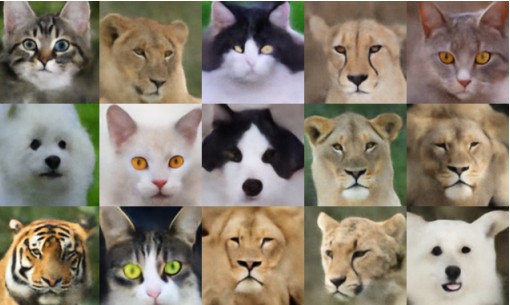

Figure 6: Examples of generated samples from $128 \times 128$ CelebA and AFHQ datasets using cold diffusion with blur transformation

## 5.3 Generation using other transformations

In this section, we provide a proof of concept that generation can be extended to other transformations. Specifically, we show preliminary results on inpainting, super-resolution, and *animorphosis*. Inspired by the simplicity of the degraded image distribution for the blurring routine presented in the previous section, we use degradation routines with predictable final distributions here as well.

To use the Gaussian mask transformation for generation, we modify the masking routine so the final degraded image is completely devoid of information. One might think a natural option is to send all of the images to a completely black image $x_T$, but this would not allow for any diversity

in generation. To get around this maximally non-injective property, we instead make the mask turn all pixels to a random, solid color. This still removes all of the information from the image, but it allows us to recover different samples from the learned distribution via Algorithm 2 by starting off with different color images. More formally, a Gaussian mask $G_t = \prod_{i=1}^{t} z_{\beta_i}$ is created in a similar way as discussed in the Section 4.2, but instead of multiplying it directly to the image $x_0$, we create $x_t$ as $G_t \cdot x_0 + (1 - G_t) \cdot c$, where $c$ is an image of a randomly sampled color.

For super-resolution, the routine down-samples to a resolution of $2 \times 2$, or $4$ values in each channel. These degraded images can be represented as one-dimensional vectors, and their distribution is modeled using one Gaussian distribution. Using the same methods described for generation using blurring described above, we sample from this Gaussian-fitted distribution of the lower-dimensional degraded image space and pass this sampled point through the generation process trained on super-resolution data to create one output.

Additionally to show one can invert nearly any transformation, we include a new transformation deemed *animorphosis*, where we iteratively transform a human face from CelebA to an animal face from AFHQ. Though we chose CelebA and AFHQ for our experimentation, in principle such interpolation can be done for any two initial data distributions.

More formally, given an image $x$ and a random image $z$ sampled from the AFHQ manifold, $x_t$ can be written as $x_t = \sqrt{\alpha_t} x + \sqrt{1 - \alpha_t} z$. Note this is essentially the same as the noising procedure, but instead of adding noise we are adding a progressively higher weighted AFHQ image. In order to sample from the learned distribution, we sample a random image of an animal and use TACoS.

We present results for the CelebA dataset, and hence the quantitative results in terms of FID scores for inpainting, super-resolution and *animorphosis* are 90.14, 92.91 and 48.51 respectively. We further show some qualitative samples in Figure 7, and in Figure 1.

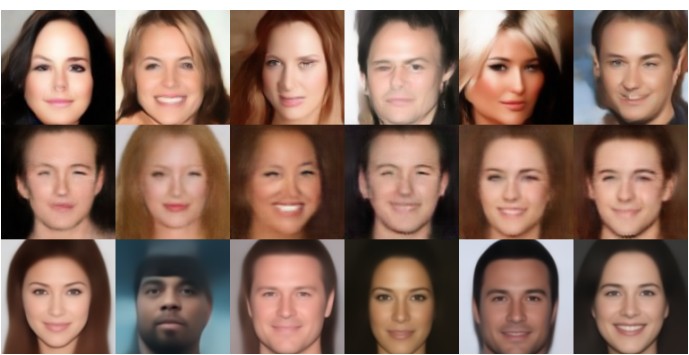

Figure 7: Preliminary demonstration of the generative abilities of other cold diffusins on the $128 \times 128$ CelebA dataset. The top row is with *animorphosis* models, the middle row is with inpainting models, and the bottom row exhibits super-resolution models.

## 6 Conclusion

Existing diffusion models rely on Gaussian noise for both forward and reverse processes. In this work, we find that the random noise can be removed entirely from the diffusion model framework, and replaced with arbitrary transforms. In doing so, our generalization of diffusion models and their sampling procedures allows us to restore images afflicted by deterministic degradations such as blur, inpainting and downsampling. This framework paves the way for a more diverse landscape of diffusion models beyond the Gaussian noise paradigm. The different properties of these diffusions may prove useful for a range of applications, including image generation and beyond.

## 7 Acknowledgements

This work was made possible by the ONR MURI program, the Office of Naval Research (N000142112557), and the AFOSR MURI program. Commercial support was provided by Capital One Bank, the Amazon Research Award program, and Open Philanthropy. Further support was provided by the National Science Foundation (IIS-2212182), and by the NSF TRAILS Institute (2229885).

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

# A Appendix

## A.1 Deblurring

For the deblurring experiments, we train the models on different datasets for 700,000 gradient steps. We use the Adam Kingma and Ba [2014] optimizer with learning rate $2 \times 10^{-5}$. The training was done on the batch size of 32, and we accumulate the gradients every 2 steps. Our final model is an Exponential Moving Average of the trained model with decay rate 0.995 which is updated after every 10 gradient steps.

For the MNIST dataset, we blur recursively 40 times, with a discrete Gaussian kernel of size 11x11 and a standard deviation 7. In the case of CIFAR-10, we recursively blur with a Gaussian kernel of fixed size 11x11, but at each step $t$, the standard deviation of the Gaussian kernel is given by $0.01 * t + 0.35$. The blur routine for CelebA dataset involves blurring images with a Gaussian kernel of 15x15 and the standard deviation of the Gaussian kernel grows exponentially with time $t$ at the rate of 0.01.

Figure 8 shows an additional nine images for each of MNIST, CIFAR-10 and CelebA. Figures 19 and 20 show the iterative sampling process using a deblurring model for ten example images from each dataset. We further show 400 random images to demonstrate the qualitative results in the Figure 21.

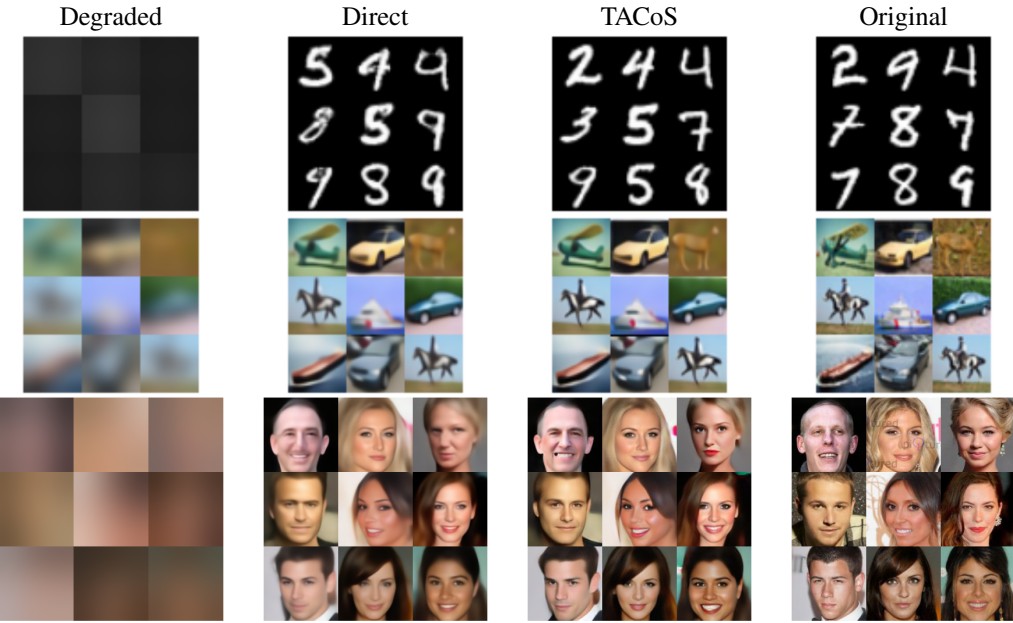

Figure 8: Additional examples from deblurring models trained on the MNIST, CIFAR-10, and CelebA datasets. **Left to right:** degraded inputs $D(x_0, T)$, direct reconstruction $R(D(x_0, T))$, sampled reconstruction with TACoS described in Algorithm 2, and original image.

## A.2 Inpainting

For the inpainting transformation, models were trained on different datasets with 60,000 gradient steps. The models were trained using Adam Kingma and Ba [2014] optimizer with learning rate $2 \times 10^{-5}$. We use batch size 64, and the gradients are accumulated after every 2 steps. The final model is an Exponential Moving Average of the trained model with decay rate 0.995. This EMA model is updated after every 10 gradient steps. For all our inpainting experiments we use a randomized Gaussian mask and $T = 50$ with $\beta_1 = 1$ and $\beta_{i+1} = \beta_i + 0.1$.

To avoid potential leakage of information due to floating point computation of the Gaussian mask, we discretize the masked image before passing it through the inpainting model. This was done by rounding all pixel values to the eight most significant digits.

Figure 10 shows nine additional inpainting examples on each of the MNIST, CIFAR-10, and CelebA datasets. Figure 9 demonstrates an example of the iterative sampling process of an inpainting model for one image in each dataset.

## A.3 Super-Resolution

We train the super-resolution model per Section 3.1 for 700,000 iterations. We use the Adam Kingma and Ba [2014] optimizer with learning rate $2 \times 10^{-5}$. The batch size is 32, and we accumulate the gradients every 2 steps. Our final model is an Exponential Moving Average of the trained model with decay rate 0.995. We update the EMA model every 10 gradient steps.

The number of time-steps depends on the size of the input image and the final image. For MNIST and for CIFAR10, the number of time steps is 3, as it takes three steps of halving the resolution to reduce the initial image down to $4 \times 4$. For CelebA, the number of time steps is 6 to reduce the initial image down to $2 \times 2$. For CIFAR10, we apply random crop and random horizontal flip for regularization.

Figure 12 shows an additional nine super-resolution examples on each of the MNIST, CIFAR-10, and CelebA datasets. Figure 11 shows one example of the progressive increase in resolution achieved with the sampling process using a super-resolution model for each dataset.

## A.4 Colorization

Here we provide results for the additional task of colorization. Starting with the original RGB-image $x_0$, we realize colorization by iteratively desaturating for $T$ steps until the final image $x_T$ is a fully gray-scale image. We use a series of three-channel $1 \times 1$ convolution filters $\mathbf{z}(\alpha) = \{z^1(\alpha), z^2(\alpha), z^3(\alpha)\}$ with the form

$$z^1(\alpha) = \alpha \left( \tfrac{1}{3} \tfrac{1}{3} \tfrac{1}{3} \right) + (1 - \alpha) \left( 1\ 0\ 0 \right)$$

$$z^2(\alpha) = \alpha \left( \tfrac{1}{3} \tfrac{1}{3} \tfrac{1}{3} \right) + (1 - \alpha) \left( 0\ 1\ 0 \right)$$

$$z^3(\alpha) = \alpha \left( \tfrac{1}{3} \tfrac{1}{3} \tfrac{1}{3} \right) + (1 - \alpha) \left( 0\ 0\ 1 \right)$$

and obtain $D(x, t) = \mathbf{z}(\alpha_t) * x$ via a schedule defined as $\alpha_1, \dots, \alpha_t$ for each respective step. Notice that a gray image is obtained when $x_T = \mathbf{z}(1) * x_0$.

We can tune the ratio $\alpha_t$ to control the amount of information removed in each step. For our experiment, we schedule the ratio such that for every $t$ we have

$$x_t = \mathbf{z}(\alpha_t) * \dots * \mathbf{z}(\alpha_1) * x_0 = \mathbf{z}(\tfrac{t}{T}) * x_0.$$

This schedule ensures that color information lost between steps is smaller in earlier stage of the diffusion and becomes larger as $t$ increases.

We train the models on different datasets for 700,000 gradient steps. We use Adam Kingma and Ba [2014] optimizer with learning rate $2 \times 10^{-5}$. We use batch size 32, and we accumulate the gradients every 2 steps. Our final model is an exponential moving average of the trained model with decay rate 0.995. We update the EMA model every 10 gradient steps. For CIFAR-10 we use $T = 50$ and for CelebA we use $T = 20$.

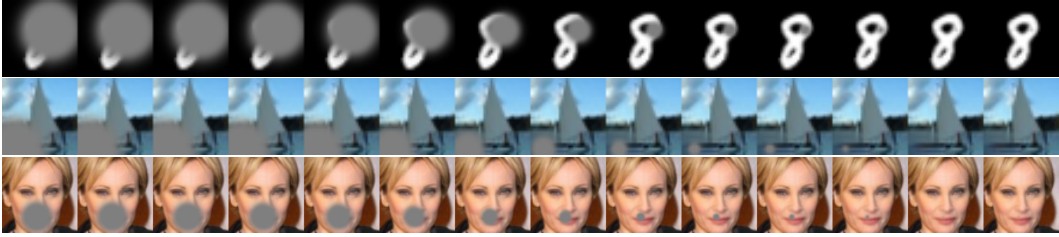

Figure 9: Progressive inpainting of selected masked MNIST, CIFAR-10, and CelebA images.

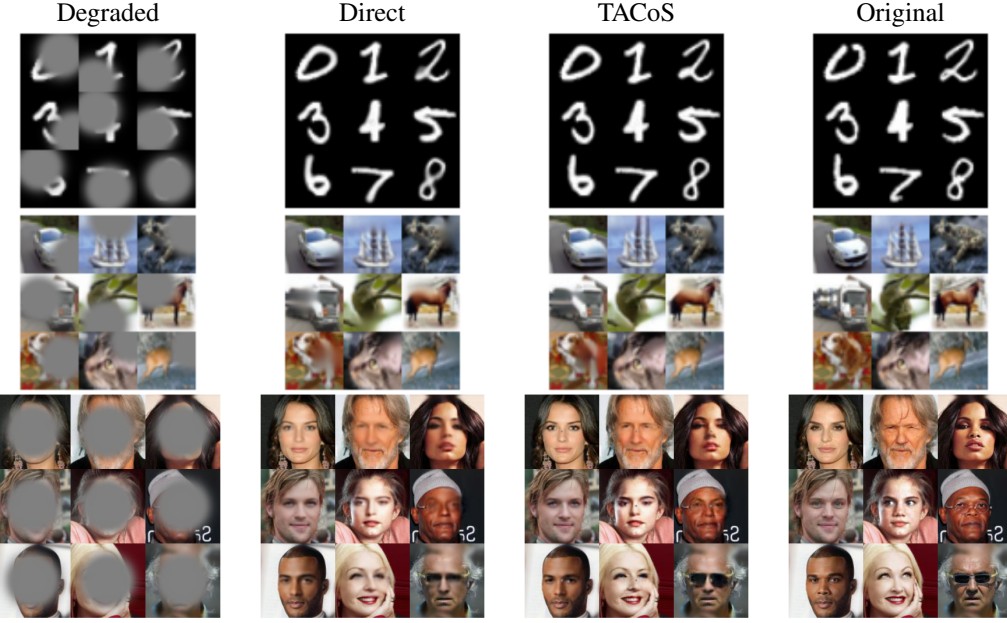

Figure 10: Additional examples from inpainting models trained on the MNIST, CIFAR-10, and CelebA datasets. **Left to right:** degraded inputs $D(x_0, T)$, direct reconstruction $R(D(x_0, T))$, sampled reconstruction with TACoS described in Algorithm 2, and original image.

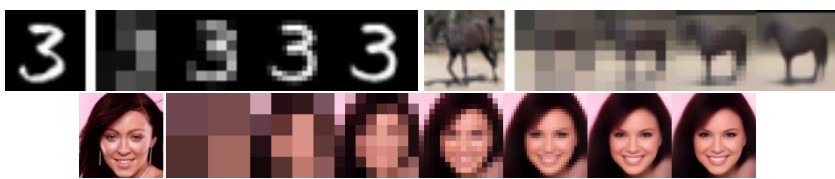

Figure 11: Progressive upsampling of selected downsampled MNIST, CIFAR-10, and CelebA images. The original image is at the left for each of these progressive upsamplings.

We illustrate our recolorization results in Figure 13. We present testing examples, as well as their grey scale images, from all the datasets, and compare the recolorization results with the original images. The recolored images feature correct color separation between different regions, and feature various and yet semantically correct colorization of objects. Our sampling technique still yields minor differences in comparison to the direct reconstruction, although the change is not visually apparent. We attribute this to the shape restriction of colorization task, as human perception is rather insensitive to minor color change. We also provide quantitative measurement for the effectiveness of our recolorization results in terms of different similarity metrics, and summarize the results in Table 5.

Table 5: Quantitative metrics for quality of image reconstruction using recolorization models for all three channel datasets.

| Dataset | Degraded Image | | | Reconstruction | | |
|---|---|---|---|---|---|---|
| | FID | SSIM | RMSE | FID | SSIM | RMSE |
| CIFAR-10 | 97.39 | 0.937 | 0.078 | 45.74 | 0.942 | 0.069 |
| CelebA | 41.20 | 0.942 | 0.089 | 17.50 | 0.973 | 0.042 |

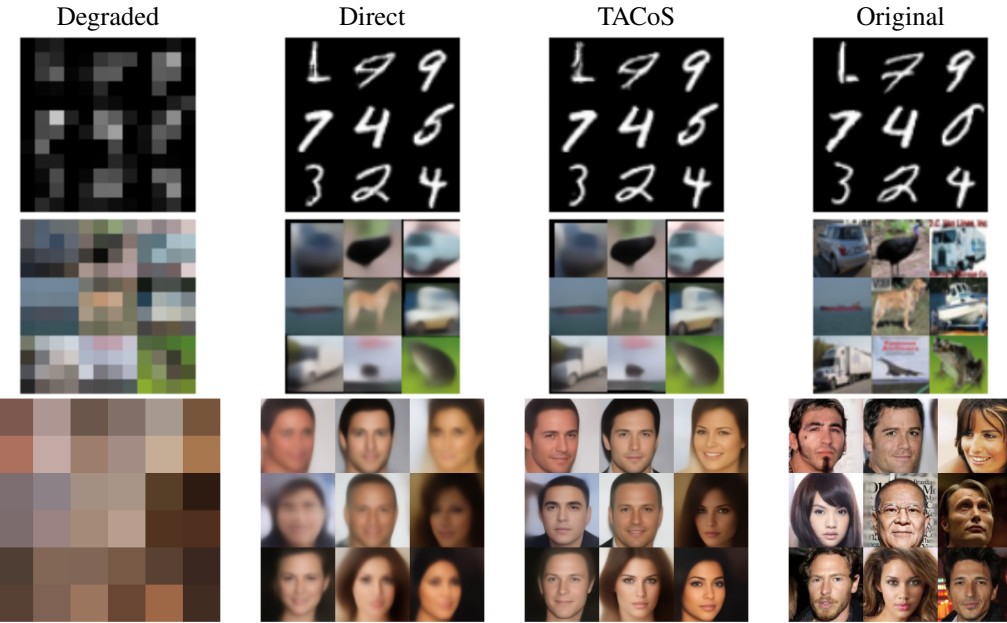

Figure 12: Additional examples from super-resolution models trained on the MNIST, CIFAR-10, and CelebA datasets. **Left to right:** degraded inputs $D(x_0, T)$, direct reconstruction $R(D(x_0, T))$, sampled reconstruction with TACoS described in Algorithm 2, and original image.

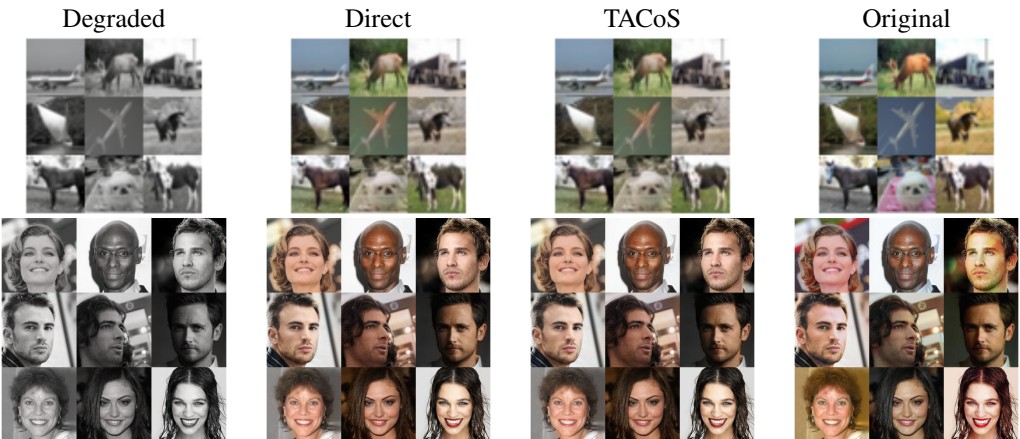

Figure 13: Recolorization models trained on the CIFAR-10 and CelebA datasets. **Left to right:** degraded inputs $D(x_0, T)$, direct reconstruction $R(D(x_0, T))$, sampled reconstruction with TACoS described in Algorithm 2, and original image.

## A.5 Image Snow

Here we provide results for the additional task of snowification, which is a direct adaptation of the offical implementation of ImageNet-C snowification process Hendrycks and Dietterich [2019]. To determine the snow pattern of a given image $x_0 \in \mathbb{R}^{C \times H \times W}$, we first construct a seed matrix $S_A \in \mathbb{R}^{H \times W}$ where each entry is sampled from a Gaussian distribution $N(\mu, \sigma)$. The upper-left corner of $S_A$ is then zoomed into another matrix $S_B \in \mathbb{R}^{H \times W}$ with spline interpolation. Next, we create a new matrix $S_C$ by filtering each value of $S_B$ with a given threshold $c_1$ as

$$S_C[i][j] = \begin{cases} 0, & S_B[i][j] \le c_1 \\ S_B[i][j], & S_B[i][j] > c_1 \end{cases}$$

and clip each entry of $S_C$ into the range $[0, 1]$. We then convolve $S_C$ using a motion blur kernel with standard deviation $c_2$ to create the snow pattern $S$ and its up-side-down rotation $S'$. The direction of the motional blur kernel is randomly chosen as either vertical or horizontal. The final snow image is created by again clipping each value of $x_0 + S + S'$ into the range $[0, 1]$. For simplicity, we abstract the process as a function $h(x_0, S_A, c_0, c_1)$.

| Degraded | Direct | TACoS | Original |
|:---:|:---:|:---:|:---:|

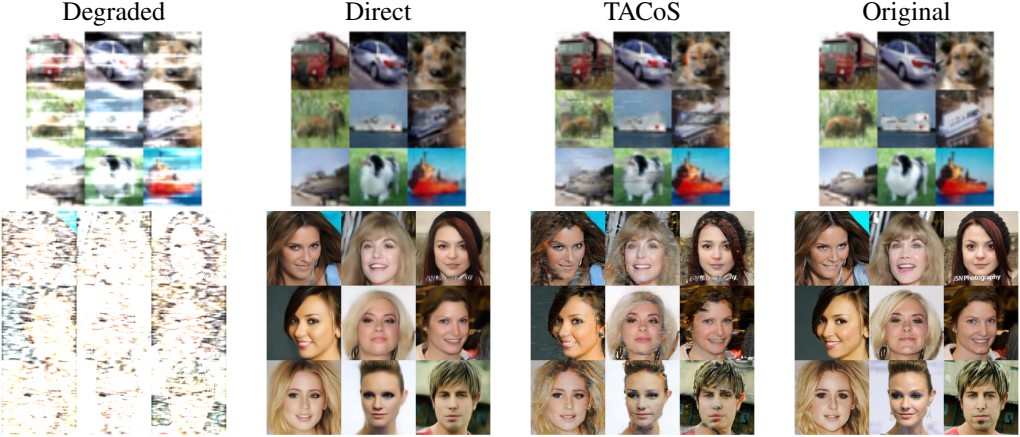

Figure 14: Additional examples from *Desnowification* models trained on the CIFAR-10 and CelebA datasets. **Left to right:** degraded inputs $D(x_0, T)$, direct reconstruction $R(D(x_0, T))$, sampled reconstruction with TACoS described in Algorithm 2, and original image.

To create a series of $T$ images with increasing snowification, we linearly interpolate $c_0$ and $c_1$ between $[c_0^{\text{start}}, c_0^{\text{end}}]$ and $[c_1^{\text{start}}, c_1^{\text{end}}]$ respectively, to create $c_0(t)$ and $c_1(t)$, $t = 1, \ldots, T$. Then for each $x_0$, a seed matrix $S_x$ is sampled, the motion blur direction is randomized, and we construct each related $x_t$ by $x_t = h(x_0, S_x, c_0(t), c_1(t))$. Visually, $c_0(t)$ dictates the severity of the snow, while $c_1(t)$ determines how "windy" the snowified image seems.

For both CIFAR-10 and Celeb-A, we use the same Gaussian distribution with parameters $\mu = 0.55$ and $\sigma = 0.3$ to generate the seed matrix. For CIFAR-10, we choose $c_0^{\text{start}} = 1.15$, $c_0^{\text{end}} = 0.7$, $c_1^{\text{start}} = 0.05$ and $c_1^{\text{end}} = 16$, which generates a visually lighter snow. For Celeb-A, we choose $c_0^{\text{start}} = 1.15$, $c_0^{\text{end}} = 0.55$, $c_1^{\text{start}} = 0.05$ and $c_1^{\text{end}} = 20$, which generates a visually heavier snow.

We train the models on different datasets for 700,000 gradient steps. We use Adam Kingma and Ba [2014] optimizer with learning rate $2 \times 10^{-5}$. We use batch size 32, and we accumulate the gradients every 2 steps. Our final model is an exponential moving average of the trained model with decay rate 0.995. We update the EMA model every 10 gradient steps. For CIFAR-10 we use $T = 200$ and for CelebA we use $T = 200$. We note that the seed matrix is resampled for each individual training batch, and hence the snow pattern varies across the training stage.

### A.6  Generation using noise : Further Details

Here we show the equivalence between the sampling method proposed in Algorithm 2 and the deterministic sampling in DDIM Song et al. [2021a]. Given the image $x_t$ at step $t$, we have the restored clean image $\hat{x}_0$ from the diffusion model. Hence given the estimated $\hat{x}_0$ and $x_t$, we can estimate the noise $z(x_t, t)$ (or $\hat{z}$) as

$$z(x_t, t) = \frac{x_t - \sqrt{\alpha_t}\hat{x}_0}{\sqrt{1 - \alpha_t}},$$

Thus, the $D(\hat{x}_0, t)$ and $D(\hat{x}_0, t - 1)$ can be written as

$$D(\hat{x}_0, t) = \sqrt{\alpha_t}\hat{x}_0 + \sqrt{1 - \alpha_t}\hat{z},$$

$$D(\hat{x}_0, t - 1) = \sqrt{\alpha_{t-1}}\hat{x}_0 + \sqrt{1 - \alpha_{t-1}}\hat{z},$$

using which the sampling process in Algorithm 2 to estimate $x_{t-1}$ can be written as,

$$x_{t-1} = x_t - D(\hat{x}_0, t) + D(\hat{x}_0, t-1)$$
$$= x_t - (\sqrt{\alpha_t}\hat{x}_0 + \sqrt{1-\alpha_t}\hat{z}) + (\sqrt{\alpha_{t-1}}\hat{x}_0 + \sqrt{1-\alpha_{t-1}}\hat{z})$$
$$= \sqrt{\alpha_{t-1}}\hat{x}_0 + \sqrt{1-\alpha_{t-1}}\hat{z}$$

$$(1)$$

which is same as the sampling method as described in Song et al. [2021a]. The only difference from the original Song et al. [2021a] is the order for estimating $\hat{x}_0$ and $\hat{z}$. The original Song et al. [2021a] paper estimated $\hat{z}$ first and then used this to predict clean image $\hat{x}_0$, while we first predict the clean image $\hat{x}_0$ and then estimate the noise $\hat{z}$.

## A.7 Generation using blur transformation: Further Details

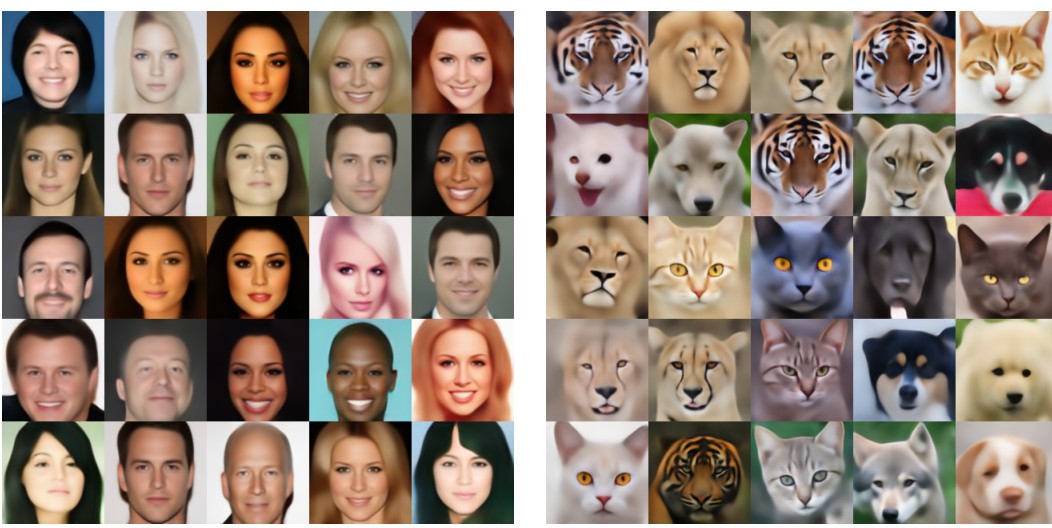

Figure 15: Examples of generated samples from $128 \times 128$ CelebA and AFHQ datasets using Method 2 with perfect symmetry.

Figure 15 shows that generation without breaking any symmetry within each channel is quite promising as well.

**Necessity of Algorithm 2:** In the case of unconditional generation, we observe a marked superiority in quality of the sampled reconstruction using Algorithm 2 over any other method considered. For example, in the broken symmetry case, the FID of the directly reconstructed images is 257.69 for CelebA and 214.24 for AFHQ, which are far worse than the scores of 49.45 and 54.68 from Table 4. In Figure 17, we also give a qualitative comparison of this difference. We can also clearly see from Figure 18 that Algorithm 1, the method used in Song et al. [2021b] and Ho et al. [2020], completely fails to produce an image close to the target data distribution.

## A.8 Algorithm 1 is same as DDIM/DDPM sampling

The sampling method proposed in Song et al. [2021a] in it's equation 12 is given as

$$x_{t-1} = \sqrt{\alpha_{t-1}} \cdot \text{``predicted } x_0\text{''} + \sqrt{1-\alpha_{t-1}-\sigma_t^2}\epsilon_\theta(x_t) + \sigma_t\epsilon_t$$

where $\epsilon_\theta(x_t)$ is the noise predicted by the diffusion model given $x_t$ and $t$. The term "predicted $x_0$" or $\hat{x}_0$ can be computed directly given $x_t$ and $\epsilon_\theta(x_t)$ as

$$\hat{x}_0 = \frac{x_t - \sqrt{1-\alpha_t}\epsilon_\theta(x_t)}{\sqrt{\alpha_t}},$$

Hence using $\hat{z}$ instead of $\epsilon_\theta(x_t)$ and $\hat{x}_0$ to indicate predicted clean image, we have

$$x_{t-1} = \sqrt{\alpha_{t-1}} \cdot \hat{x}_0 + \sqrt{1 - \alpha_{t-1} - \sigma_t^2}\hat{z} + \sigma_t\epsilon_t$$

Thus, the sampling step can interpreted as follows: At each step $t$, we start with a noisy image $x_t$ and use the diffusion model to estimate the clean image $\hat{x}_0$ and the noise $\hat{z}$ that was added to this clean image $\hat{x}_0$ to get the noisy image $x_t$. In order to move to lesser noisy image $x_{t-1}$, one "adds back" lesser noise to the the "predicted clean image" $\hat{x}_0$. Now one can add back noise in 2 ways, either the noise which was added to the clean image $\hat{x}_0$ which is $\hat{z}$ or sample a new uncorrelated noise $\epsilon_t$. Infact both of these noise can be added using $\sigma_t$ as the hyperparameter that weighs the amount of each noise added. This $\sigma_t$ is placed in the equation such that for any choice of $\sigma_t$, the standard deviation of noise added back is $\sqrt{1 - \alpha_{t-1}}$. For $\sigma_t = 0$, we only add back the estimated noise $\hat{z}$ and no uncorrelated noise $\epsilon_t$ which is infact the DDIM sampling. While for $\sigma_t = \sqrt{(1 - \alpha_{t-1})/(1 - \alpha_t)}\sqrt{1 - \alpha_t/\alpha_{t-1}}$ we get the sampling method described in DDPM.

Nevertheless, for any choice of $\sigma_t$, the sampling method involves a denoising operation which is shown as $R(x_s, s)$ in Algorithm 1 and adding back noise shown as $x_{s-1} = D(\hat{x}_0, s-1)$ in Algorithm 1. The only difference between different sampling methods explained in DDPM or DDIM is how one degrades the image back.

### A.9 Discussion on Algorithm 2 produces lesser error than Algorithm 1

Consider the case where the degradation is a simple blur operator that removes one frequency from an image every time $t$ increases by one. We can write $X = x_0 + x_1 \ldots x_{T-1} + x_T$, where each $x_i$ is the Fourier mode of $X$ representing frequency $T - i$. The degradation operator for this blur is $D(X, t) = x_t + \ldots x_{T-1} + x_T = \sum_{i=t}^{T} x_i$, and when $t = T$ the signal has been blurred into a constant vector.

Suppose we are performing the reverse process, and we begin at step $t$ with $X_t = x_t + \ldots x_T$. Now in order to go from $X_t$ to $X_{t-1}$, we first use the diffusion model to predict $\hat{X} = R(X_t, t)$. This $\hat{X}$ has an analogous Fourier expansion $\hat{X} = \sum_{n=0}^{T} \hat{x}_n$, where $\hat{x}_0$ is the highest frequency mode and $\hat{x}_T$ is the DC mode. At step $t$, the error $E_t$ can be defined as $\|X_{t-1} - \hat{X}_{t-1}\|$.

For the Algorithm 1, in which $X_{t-1}$ is given by $\hat{X}_{t-1} = D(\hat{X}, t-1)$, we can expand the error as follows

$$E_t^2 = \|X_{t-1} - \hat{X}_{t-1}\|^2$$
$$= \|\sum_{i=t-1}^{T} x_i - \sum_{i=t-1}^{T} \hat{x}_i\|^2$$
$$= \|\sum_{i=t-1}^{T} (x_i - \hat{x}_i)\|^2$$
$$= \sum_{i=t-1}^{T} \|x_i - \hat{x}_i\|^2$$

For the Algorithm 2, in which $X_{t-1}$ is given by $\hat{X}_{t-1} = X_t - D(\hat{X}, t) + D(\hat{X}, t-1)$, we can expand the error as follows

$$E_t^2 = \|X_{t-1} - (X_t - D(\hat{X}, t) + D(\hat{X}, t-1))\|^2$$
$$= \|(D(\hat{X}, t) - D(\hat{X}, t-1)) - (X_t - X_{t-1})\|^2$$
$$= \|(\sum_{i=t}^{T} \hat{x}_i - \sum_{i=t-1}^{T} \hat{x}_i) - (\sum_{i=t}^{T} x_i - \sum_{i=t-1}^{T} x_i)\|^2$$
$$= \|\hat{x}_{t-1} - x_{t-1}\|^2$$

Hence, we can see that the error incurred at step $t$ using algorithm 1 is $\sum_{i=t-1}^{T} \|x_i - \hat{x}_i\|^2$ which is clearly greater than or equal to error incurred by using the algorithm 2 which is $\|\hat{x}_{t-1} - x_{t-1}\|^2$. We

now further break down the analysis based on different scenarios depending on the quality of the reconstruction operator $R$:

1. **R is perfect.**
   In this scenario, both Algorithm 1 and Algorithm 2 are indistinguishable, and the error is 0 for both.

2. **R is imperfect only for $x_{t-1}$ and reconstructs other signals perfectly.**
   In this scenario, both Algorithm 1 and Algorithm 2 incur the same error, which is $(x_{t-1} - \hat{x}_{t-1})^2$.

3. **R is imperfect for more than one frequency**
   This is the realistic scenario. In this case, the error incurred by Algorithm 1 is strictly greater than Algorithm 2.

Thus we prove that for the realistic scenario i.e. when R is not a perfect reconstruction operator, the error incurred using Algorithm 1 is always greater than the Algorithm 2.

### A.10   Empirical comparisons between Algorithm 2 and Algorithm 1

For all the degradations in Section 4, we compare to "Direct Sampling" which is in fact the "one step" reconstruction of the input to Algorithm 2. One of the main contributions of our work is Algorithm 2, which outperforms Algorithm 1 across all degradations. Infact the Algorithm 1 is worse than the one-step generation as well. To demonstrate this concretely we present the FID results in 6, where we can clearly see that FID scores for Algorithm 1 are worse than both the one-step and Algorithm 2. We also show qualitatively how bad the Algorithm 1 is in Figures 16 and 18. In fact, for the case of the celebA dataset, Algorithm 1 fails drastically, while for CIFAR-10 we can see high-frequency signals present in the image generation. Hence, we chose our baseline to be direct generation instead of Algorithm 1.

Table 6: FIDs for blur degradation for Algorithm 1, Algorithm 2 and Direct Reconstruction. This table demonstrates that Algorithm 1 is worse than both the one-step generation and Algorithm 2

| Dataset | Direct Generation | Algorithm 1 | Algorithm 2 |
|---------|-------------------|-------------|-------------|
| MNIST | 5.10 | 8.24 | 4.69 |
| CIFAR-10 | 83.69 | 97.89 | 80.08 |
| CelebA | 36.37 | 299.61 | 26.14 |

| Algorithm 1 | Algorithm 2 |
|:---:|:---:|

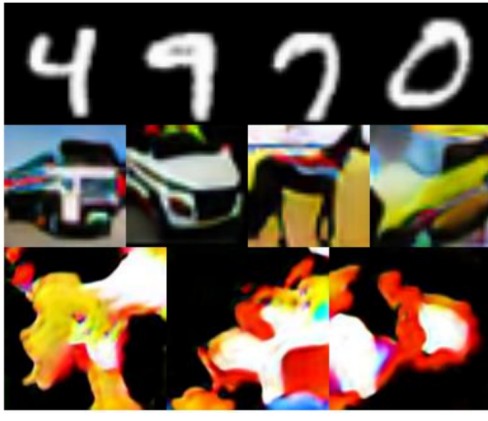 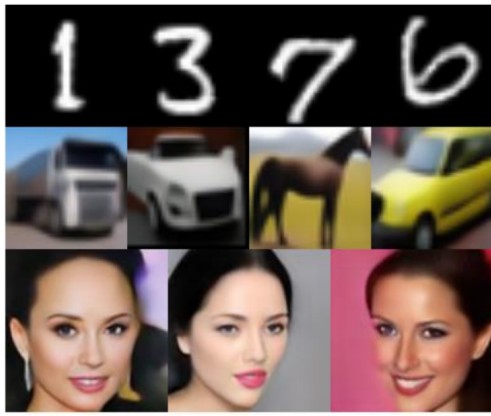

Figure 16: Comparison of Algorithm 1 and Algorithm 2. We demonstrate that Algorithm 1 performs much worse than Algorithm 1 and fails completely for CelebA dataset.

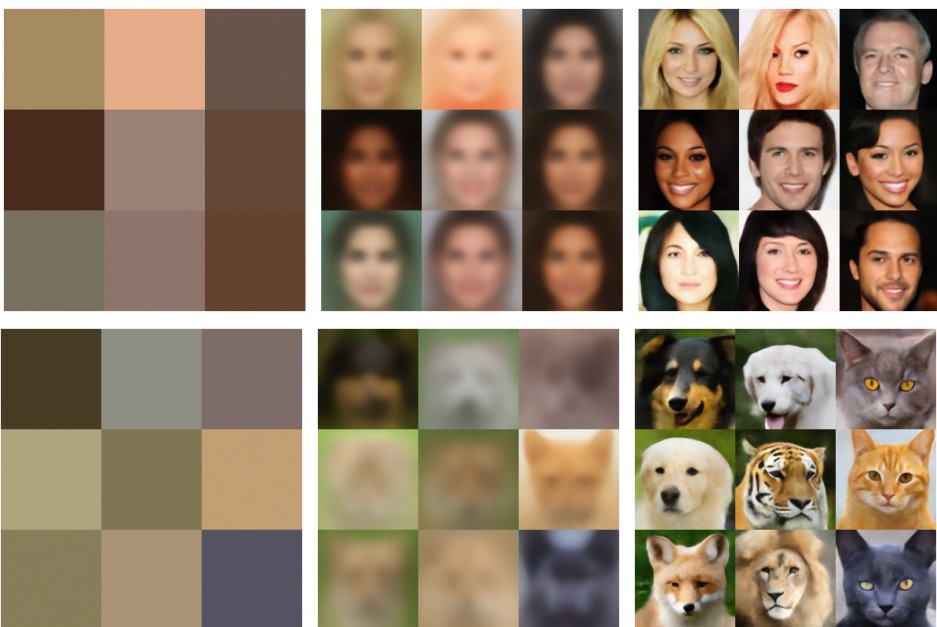

Figure 17: Comparison of direct reconstruction with sampling using TACoS described in Algorithm 2 for generation with blur transformation and broken symmetry. Left-hand column is the initial cold images generated using the simple Gaussian model. Middle column has images generated in one step (*i.e.* direct reconstruction). Right-hand column are the images sampled with TACoS described in Algorithm 2. We present results for both CelebA (top) and AFHQ (bottom) with resolution $128 \times 128$.

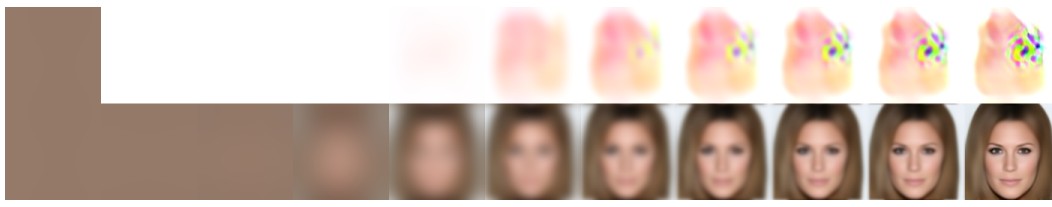

Figure 18: Comparison of Algorithm 1 (top row) and Algorithm 2 (bottom row) for generation with Method 2 and broken symmetry on $128 \times 128$ CelebA dataset. We demonstrate that Algorithm 1 fails completely to generate a new image.

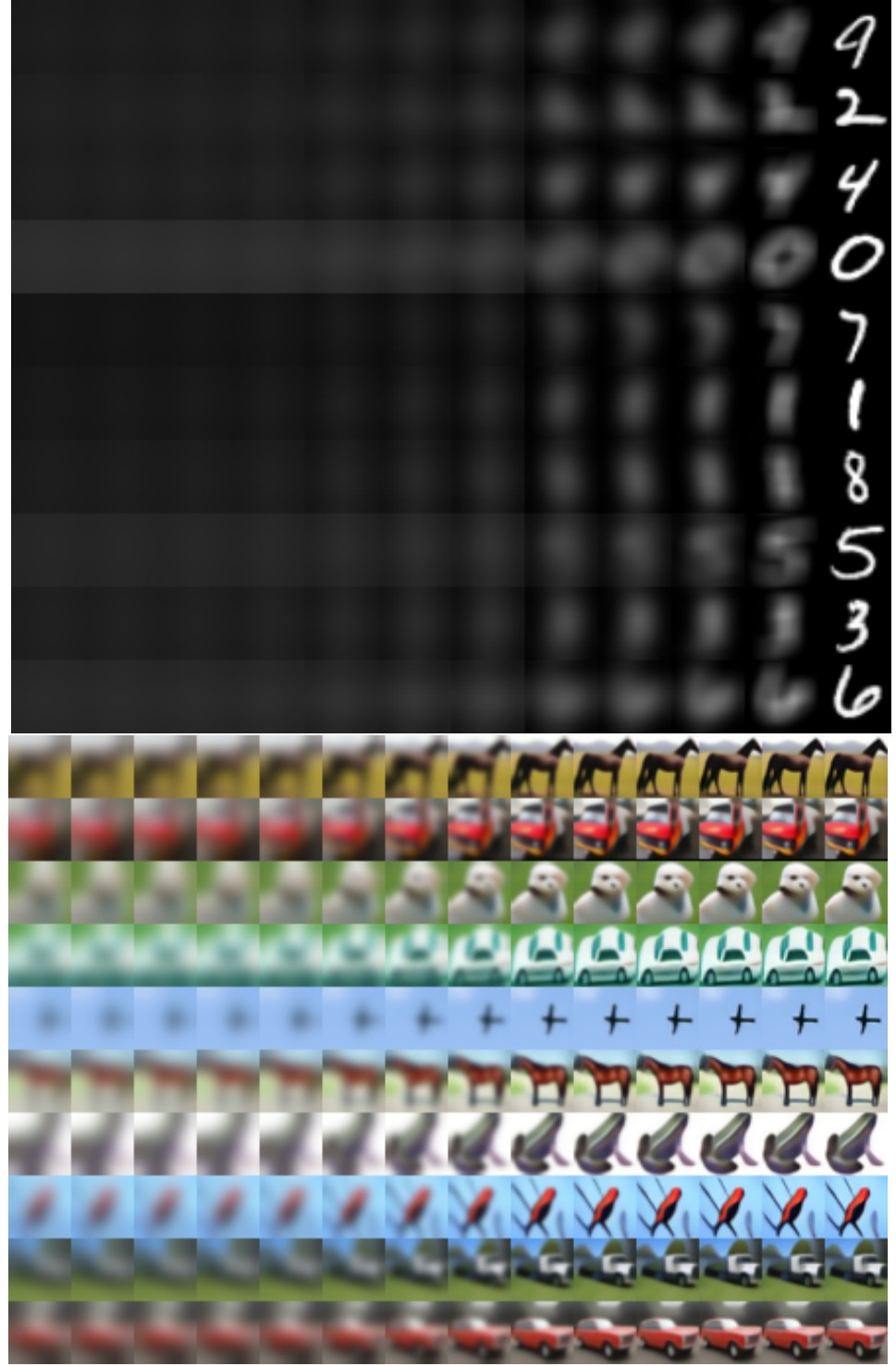

Figure 19: Progressive deblurring of selected blurred MNIST and CIFAR-10 images.

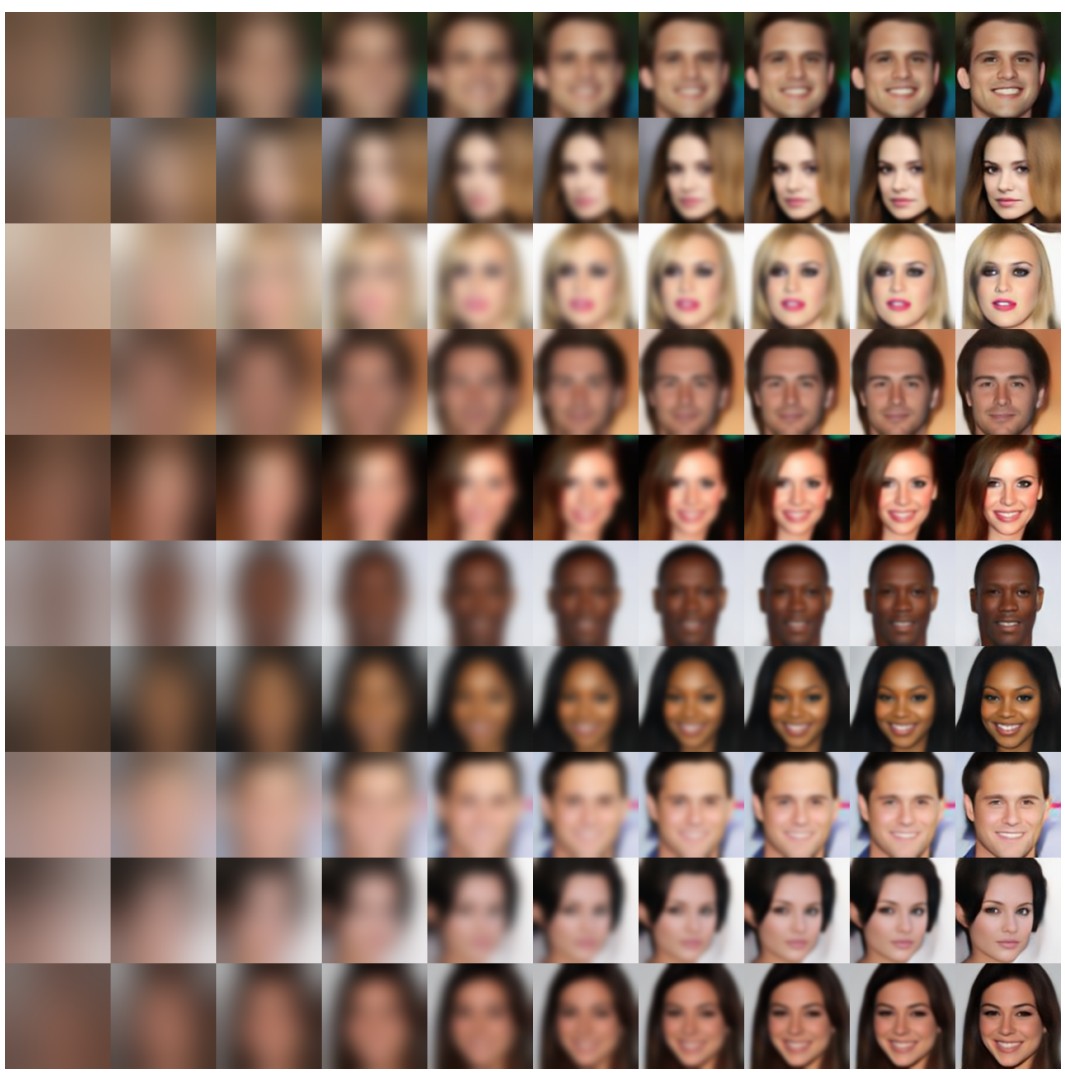

Figure 20: Progressive deblurring of selected blurred CelebA images.

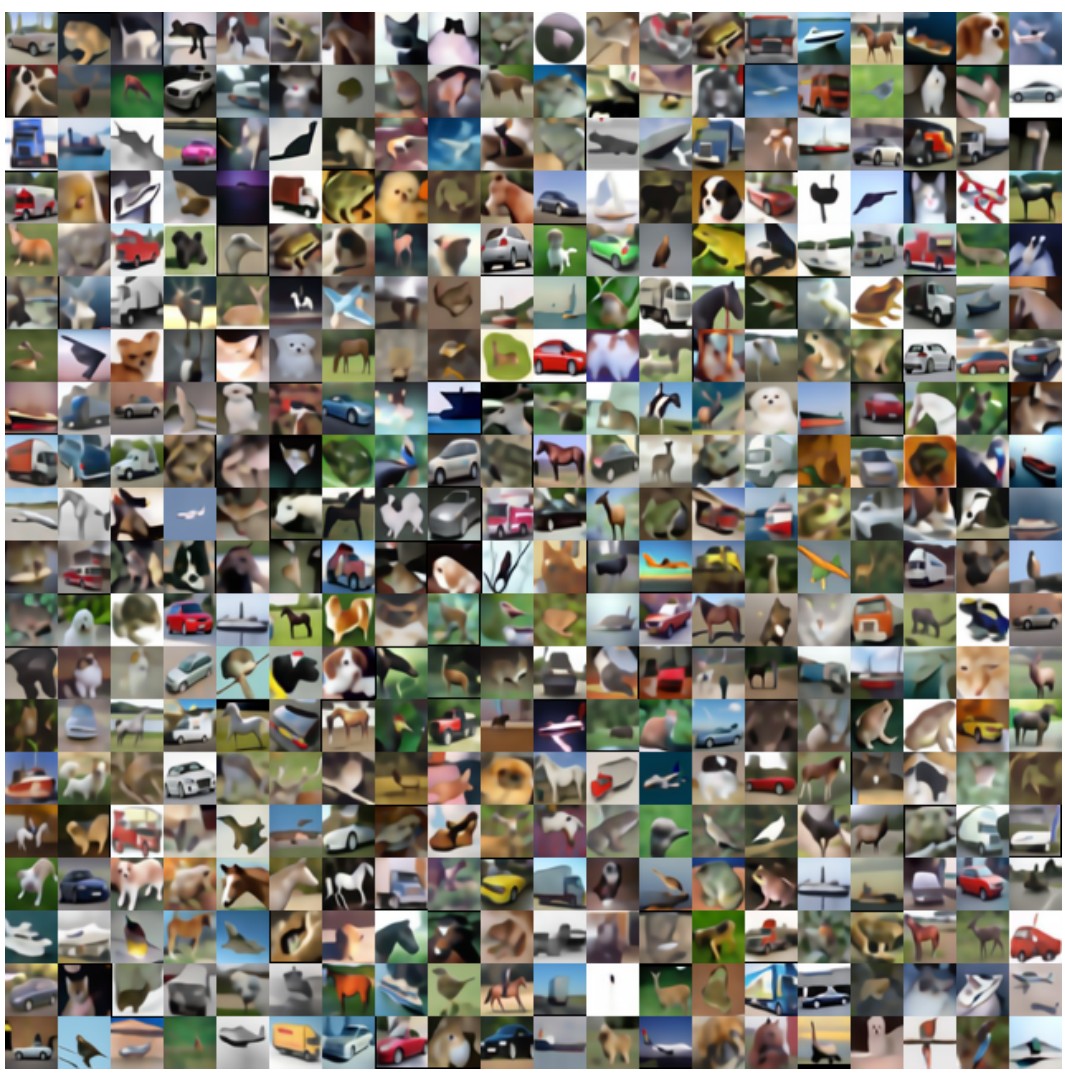

Figure 21: Deblurred Cifar10 images

