# OpenReview forum: "Cold Diffusion: Inverting Arbitrary Image Transforms Without Noise"
_NeurIPS.cc/2023/Conference — NeurIPS 2023 poster_

### Official Review · Reviewer_pjEM · 2023-07-03

**Soundness:** 2 fair
**Presentation:** 2 fair
**Contribution:** 3 good
**Rating:** 5
**Confidence:** 4

**Summary:**

The paper presents a general method for learning to de-corrupt datapoints in order to perform generative modelling. The paper proposes a sampling method that applies iterative updates to the current state that significantly outperforms a naive sampling algorithm which reconstructs and denoises alternatively. The authors present a wide variety of corruption processes and outperform a single step model and the naive algorithm.

**Strengths:**

Extending the idea of creating a generative model from corrupting data and learning to reconstruct it to a more general framework is a problem I believe many in this field are interested in. Not least for me personally, I have been interested in this problem and their sampling algorithm would have been useful for me to know when investigating this topic. This paper takes a very general approach making very few assumptions on the corruption process and makes a non-trivial contribution by coming up with a sampling algorithm that can perform iterative updates to generate new data which is found to be crucial for this type of method to work well.

The paper and methodology have flaws as I discuss next but I do believe that researchers will build on this work and find this contribution useful. The line of work is more engineering focussed and less based on theory which is not necessarily wrong and I think the novelty and likely interest in the topic slightly outweighs the negatives in this case.

**Weaknesses:**

Starting out, the performance of the method is just not that good in terms of sample quality. I appreciate that getting methods to work well takes development over the course of multiple papers but I worry that this is a limitation of the method itself due to most corruptions tested not working well compared to standard diffusion models.

When using this framework as a generative model there appear to be issues when the deterministic corruption transformation is not 1-1 due to the corrupted space being much smaller e.g. the space of images with constant colour or completely blurred images. The authors report problems with reduced diversity because of this and need to add a little bit of noise to increase diversity. I think this is quite a major flaw in the design because unlike standard diffusion models that can rely on maximum likelihood arguments to enable coverage of the data distribution, there are no such guarantees here and it is not clear how much the noise trick alleviates the issue. This flaw seems quite fundamental to some corruption process and a proper investigation into the diversity of samples would be good. Its a little strange how in effect all of the data distribution is being squeezed into this small subspace and then perturbing around the subspace a little bit induces diversity, this seems to be quite ill-conditioned in how small changes in input make large changes in generated output.

The claims about new questions being raised as to the necessity of noise in generation should be reigned in because of the popularity and performance of flow matching type methods that are deterministic during training (given a randomly sampled pair) and sampling. The links to these methods should also be discussed especially in the case of corruptions such as animorphosis which is quite similar to a flow between two arbitrary data distributions and building a interpolating bridge between two random samples from them.

The sections on deblurring and inpainting should be better introduced as the main paper talks about generative models and so this section comes as a bit of a surprise when I would have expected pure unconditional generation as the initial experiment.

Typos:
 equation after line 145, need .e on final line
 proof A.9, need i = t-1 on third line of E_t^2. How do you move the sum out of the norm on the final line ?

Edit after rebuttal: I have read the author's rebuttal and as I mention in my reply,  my points regarding the ill-conditioned nature of the generative process and poor performance are still concerning for me and I intend to leave my score as it is.

**Questions:**

The inpainting results seem so much better than other results in terms of sample quality. When you say Figure 4/Figure 10 is showing 'test images' do you actually mean that these were held out during training or has the network seen those during training. This should be made very clear since it seems that the network has just memorized those images (at least on celebA).

**Limitations:**

Discussed in weaknesses section.

---

> ### Author Rebuttal · Authors · 2023-08-10
>
> > the performance of the method is just not that good in terms of sample quality.
>
> We acknowledge that the quality of images generated in Section 5 is not comparable to that of Gaussian noise as a degradation. This work is not meant to be a SOTA method paper, instead we challenge both theory frameworks where noise is a critical component for learning the training distribution and the engineering status-quo that has not made explorations beyond Gaussian noise. We believe our experiments effectively demonstrate that entirely noise-free schemes (blur, inpainting, super-resolution) can still work.
>
> > The sections on deblurring and inpainting should be better introduced as the main paper talks about generative models and so this section comes as a bit of a surprise when I would have expected pure unconditional generation as the initial experiment.
>
> We thank the reviewer for their valuable suggestion, on our end we introduced conditional generation in Section 4 before unconditional generation in Section 5 because we wanted to present the conditional generation as a first step as it occurred in our own project. If one fails to perform conditional generation, then one will fail on unconditional generation as well which is a more difficult problem.
>
> > Typos: equation after line 145, need .e on final line proof A.9, need i = t-1 on third line of E_t^2. How do you move the sum out of the norm on the final line ?
>
> We thank the reviewer for pointing us to these typos which we have now fixed in the current revision.
> In the toy problem discussed in the Appendix A.9, we consider a blur operator that removes one frequency from image every time $t$ increase by one. Hence, given a random sample X, the vevtors $x_j$ and $x_k$ for $j \neq k$ in the expansion $X = \sum_{i=0}^{T} x_i$ are orthogonal to each other. \
> Thus we have \
> $E_t ^ 2 = ||\sum_{i=t-1}^{T} (x_i - \hat x_i) || ^ 2$ \
> $E_t ^ 2 = \sum_{i=t-1}^{T} || (x_i - \hat x_i) || ^ 2 +  \mathop{\sum\sum}_{j\neq k} (x_j - \hat x_j)^T(x_k - \hat x_k)$
>
> Since both $X$ and $\hat X$ have the same orthogonal basis, i.e. the fourier directions, we have terms $x_j^Tx_k$, $x_j^T\hat x_k$, $x_k^Tx_j$ and $x_k^T\hat x_j$ to be equal to zero. This results in \
> $E_t ^ 2 = \sum_{i=t-1}^{T} || (x_i - \hat x_i) || ^ 2$
>
>
>
> > When you say Figure 4/Figure 10 is showing 'test images' do you actually mean that these were held out during training or has the network seen those during training.
>
> In all of our experiments, we evaluate our methods on the held-out testing dataset. We specify the evaluation details in lines 169-173.
>
> Thank you again for your thoughtful review. We made an effort to address your feedback including paper edits and would appreciate it if you would consider raising your score in light of our response.  Do you have any additional questions we can address?

---

> > ### Comment · Reviewer_pjEM · 2023-08-11
> >
> > Thank you for your response, I appreciate your answers to the questions. My points regarding the ill-conditioned nature of the generative process and poor performance are still concerning for me and I intend to leave my score as it is.

---

### Official Review · Reviewer_pgXS · 2023-07-04

**Soundness:** 3 good
**Presentation:** 3 good
**Contribution:** 4 excellent
**Rating:** 7
**Confidence:** 5

**Summary:**

This paper extends the Gaussian diffusion model toward arbitrary image-to-image translations, named Cold Diffusion. Specifically, the authors define a generalized forward diffusion process and its training process, then propose a novel Transformation Agnostic Cold Sampling (TACoS) process for generations.  Experiments show that Cold Diffusion can effectively achieve image generation by learning image-to-image translation.

**Strengths:**

This paper proposed a novel idea that the diffusion process can be applied to arbitrary image-to-image translations, not limited to Gaussian noise. The ideas proposed in this paper have achieved a certain impact in the field of diffusion models and inspired a lot of work. In light of this, I recommend that the paper be accepted.

**Weaknesses:**

The author only provides an empirical formulation, without rigorous theoretical analysis. Nonetheless, this cannot overshadow the novelty of this work.

**Questions:**

Is it possible to apply the diffusion process to any domain translation, i.e., dimension agnostic and modality agnostic?

**Limitations:**

1. It seems that the image-to-image translation needs to have invariant dimensions.
2. It will be interesting to have a mathematical analysis of Cold Diffusion.

---

> ### Author Rebuttal · Authors · 2023-08-10
>
> We deeply appreciate your careful review and positive assessment of our work. Your recognition of the novel approach and potential impact in the field of diffusion models is highly encouraging. Below, we address the points you raised:
>
> > Is it possible to apply the diffusion process to any domain translation, i.e., dimension agnostic and modality agnostic?
>
> Yes, we believe our proposed algorithm is modality agnostic and can be adapted to different modalities like speech, text, etc. though diffusion modeling as a method is difficult to apply in variable-dimension settings. For example, text can occur in various lengths and diffusion models whose corruption process can result in text of different lengths present difficulties.  Solving this problem with Cold Diffusion may or may not be more difficult.

---

> ### Comment · Reviewer_pgXS · 2023-08-16
>
> Thanks for your response. I will keep my rating.

---

### Official Review · Reviewer_VoTU · 2023-07-05

**Soundness:** 3 good
**Presentation:** 4 excellent
**Contribution:** 2 fair
**Rating:** 4
**Confidence:** 5

**Summary:**

This work introduces a novel approach called cold diffusion, in which both the forward and backward processes are deterministic. The authors propose a scheme called Tacos, which predicts x_{s-1} from x_s by leveraging the estimated increment D(\hat{x}_0, s) - D(\hat{x}_0, s-1).

**Strengths:**

The autors propsed nontrivial generalization of diffusion generative model to simple and straightforward deterministic process.


**Weaknesses:**

1. The justification of TACos (Section 3.3) appears weak.
- Higher-order terms may have a significant impact that is not adequately addressed.
- TACoS is likely to fail in standard Gaussian diffusion scenarios.

2. As mentioned in Section 5.2, the generated samples exhibit low diversity.
This indicates a failure to accurately recover the sample distribution, which is a primary objective of diffusion generative models.

3. Beyond the issue of diversity, the quality of the generated samples is also questionable.
In the appendix, the 128x128 generated samples demonstrate significantly poorer quality compared to regular generative models.

**Questions:**

The output quality of other cold generations is questionable. For instance, in the case of super-resolution, starting from a 2x2 image for the backward process may lead to similar issues of low diversity in the generated samples.

**Limitations:**

This reviewer appreciates the authors' introduction of a novel deterministic generative diffusion framework. However, it is important for the proposed framework to demonstrate comparability to reasonable GAN models in terms of diversity and output quality. The inclusion of Gaussian noise (as seen in predictor-corrector models or Langevin dynamics) or a similar randomization process is typically considered necessary and serves as a key component in diffusion processes. To justify the diffusion without noise, it is crucial for the authors to provide results that are at least comparable to decent GAN models.

---

> ### Author Rebuttal · Authors · 2023-08-10
>
> Thank you for your thoughtful review and for recognizing the nontrivial generalization of our work. We address each of your points below:
>
> > Higher-order terms may have a significant impact that is not adequately addressed.
>
> In lines 156-160, we mention that the analysis in Section 3.3 is not a complete convergence theory but rather is to highlight a desirable theoretical property of TACoS that a naive sampler lacks. In the analysis presented in lines 138-148, we assume t to be approximately 0. This implies that the difference between $D(x, t=0)$ and  $D(x, t=\epsilon)$ is sufficiently small to ignore higher-order terms, compared to the first order terms. \
> This analysis is limited to small $t$ and raises the question if this advantage of Algorithm 2 (TACoS) over the naive Algorithm 1 extends to region when t is not close to 0. To this end, we prove in Appendix A.9 that for a toy problem in which the blur operator removes one frequency at a time, the error incurred by Algorithm 1 is higher than Algorithm 2 for all $t$.
>
>
> > TACoS is likely to fail in standard Gaussian diffusion scenarios.
>
> We appreciate your concern, but as we discuss in detail in Section 5.1 and Appendix A.6, DDIM is a special case of the proposed TACoS algorithm when applied to Gaussian noise. This means that both Algorithm 1 and Algorithm 2 are equivalent to DDIM in the case of Gaussian diffusion. We explicitly verify this in Table 4, where we train a hot diffusion model, i.e. we use Gaussian noise as degradation. As standard diffusion is recovered as a special case, this indeed works well and TACoS does not fail for standard Gaussian diffusion scenarios.
>
>
> > The output quality of other cold generations is questionable. For instance, in the case of super-resolution, starting from a 2x2 image for the backward process may lead to similar issues of low diversity in the generated samples.
>
> In our experiments, we observed that the *starting point*, whether a 2x2 image is used for down-sampling based degradation in Table 3 or a 1x1 image in the case of cold diffusion in Table 5, has a distinct impact on the final results because of the nature of degradation involved. Your observation about the potential issues of low diversity in the generated samples starting from a 2x2 image is insightful. However, we found that the underlying cause of sub-par performance is related to the number of steps, not necessarily the diversity in the 2x2 region.
>
> Specifically:
>
> * In the case of blur-based cold generations (Table 4), perfect symmetry at the starting point leads to FID scores similar to those obtained when starting from a 2x2 image using downsampling as degradation (Table 3).
> * The key difference between these results lies in the sampling steps. For cold generation, we utilized 300 steps to generate 128x128 CelebA images, while in the case of downsampling-based degradation, we use just 6 steps.
> * Consequently, we believe that the limitation in performance is attributable to the fewer number of steps rather than less diversity in the 2x2 region.
> * Moreover, for downsampling-based degradation, our results demonstrate that starting from 2x2 images produces more diverse samples compared to 1x1 images for the CelebA dataset.
>
>
> > This reviewer appreciates the authors' introduction of a novel deterministic generative diffusion framework. However, it is important for the proposed framework to demonstrate comparability to reasonable GAN models in terms of diversity and output quality. The inclusion of Gaussian noise (as seen in predictor-corrector models or Langevin dynamics) or a similar randomization process is typically considered necessary and serves as a key component in diffusion processes. To justify the diffusion without noise, it is crucial for the authors to provide results that are at least comparable to decent GAN models.
>
> We thank the reviewer for their appreciation of our work on introducing a novel deterministic generative diffusion framework. In Table 4, where we compare cold diffusion to "hot diffusion" (which uses Gaussian noise as degradation), the use of estimated noise in TACoS sampling makes it equivalent to DDIM [1], a deterministic sampling method, which is in fact better than sampling methods that use noise like DDPM [2]. Hence on our end, we compare our cold diffusion with a standard and widely accepted generative model. \
> Moreover, this is not meant to be a SOTA method paper (we have updated our local draft to clarify). Instead, this work challenges both theory frameworks where noise is a **critical** component for learning the training distribution and the engineering status-quo that has explored little beyond Gaussian noise. We believe our experiments effectively demonstrate that entirely noise-free schemes (blur, inpainting, super-resolution) can still work. This will not only be of theoretical interest, but it may move the community to explore other kinds of diffusion for common tasks like upsampling, and with further research investments such methods may someday become standard tools.
>
> [1] Denoising Diffusion Implicit Models \
> [2] Denoising Diffusion Probabilistic Models
>
> Thank you again for your thoughtful review. We made an effort to address your feedback including paper edits and would appreciate it if you would consider raising your score in light of our response.  Do you have any additional questions we can address?

---

> > ### Comment · Reviewer_VoTU · 2023-08-16
> > **Clarification on my previous comment regarding Gaussian case**
> >
> > I continue to harbor doubts regarding its applicability to the Gaussian scenario, especially in the Variance Exploding setup defined by $x_t = x + W_t$ where $W_t$ is a standard Brownian motion characterized by $W_t\sim \mathcal{N}(0, t)$.
> > For this case, Algorithm 2 fundamentally operates by subtracting noise: $x_{s-1} = x_s - Z$ where $Z = N_s-N_{s-1}$ is essentially a Gaussian noise. Given this setup, I'm skeptical about the algorithm's ability to produce accurate samples.

---

> > > ### Author Response · Authors · 2023-08-17
> > > **Clarification on Variance Exploding setup for Gaussian Noise**
> > >
> > > Thank you for engaging with us in the discussion. We would like to clear that Algorithm 2 will work even for Variance Exploding setup in this discussion.
> > >
> > > For the case of VE setup we have
> > > $x_t = x_{t-1} + \sqrt{\sigma_t^2 - \sigma_{t-1}^2}\epsilon$
> > > where $\epsilon$ is sampled from $N(0, I)$ and $\sigma_t$ is such that it increases with time $t$. We use this nomenclature of VE from equation 20 of SBM [1] present in Appendix B, which shows that it results in VE SDE.
> > >
> > > This VE in discrete form can be further simplified as
> > > $x_t = x_0 + \sigma_t \epsilon$
> > >
> > > Hence, the degradation model $D(x_0, t)$ gives $x_t = x_0 + \sigma_t \epsilon$ and the reconstruction operation $R(x_t, t)$ predicts the clean image $\hat x_0$. Thus the Algorithm 2 will give us
> > > $x_{t-1} = x_t - D(\hat x_0, t) + D(\hat x_0, t-1)$
> > >
> > > As discussed in section 5.1 if one uses the *estimated noise* in degradation which in this case is $\hat \epsilon = \frac{x_t - \hat x_0}{\sigma_t}$, we have
> > >
> > > $D(\hat x_0, t) = \hat x_0 + \sigma_t \hat \epsilon$ \
> > > $D(\hat x_0, t) = \hat x_0 + \sigma_t \frac{x_t - \hat x_0}{\sigma_t}$ \
> > > $D(\hat x_0, t) = \hat x_0 + x_t - \hat x_0$ \
> > > $D(\hat x_0, t) = x_t$
> > >
> > > Thus this simplifies, the sampling in proposed algorithm 2 as  \
> > > $x_{t-1} = x_t - x_t + D(\hat x_0, t-1)$ \
> > > $x_{t-1} = D(\hat x_0, t-1)$
> > >
> > > This $D(\hat x_0, t-1)$ can be simplified as \
> > > $D(\hat x_0, t-1)= \hat x_0 + \sigma_{t-1} \hat \epsilon$
> > >
> > > Hence we get \
> > > $x_{t-1} = \hat x_0 + \sigma_{t-1} \hat \epsilon$ from our proposed algorithm 2, which in fact is the deterministic sampling algorithm for VE setup.
> > >
> > > Thank you again for your thoughtful response. We made an effort to address your feedback including paper edits and would appreciate it if you would consider raising your score in light of our response.  Do you have any additional questions we can address?

---

### Official Review · Reviewer_SFSB · 2023-07-05

**Soundness:** 3 good
**Presentation:** 3 good
**Contribution:** 2 fair
**Rating:** 4
**Confidence:** 4

**Summary:**

This paper introduces a method for image generation based on generic degradation and reconstruction operators. The approach generalizes diffusion models, which correspond to degradation by additive Gaussian noise, and reconstruction by denoising. In TACoS, the sampling scheme is agnostic to the choice of image degradation, and the corresponding restoration operator is learned via least squares regression over the data.

The authors additionally introduce a sampling iteration with a correction term (Algorithm 2) that induces first order cancellation of errors induced by improper learning of the reconstruction operator. The correction term is shown to greatly improve performance over a naive approach, since it prevents blow-up of fitting error over multiple iterations. The authors also prove that in a toy problem (degradation via frequency filtering) that Algorithm 2 has smaller reconstruction error than the naive approach.

Finally, the authors demonstrate that TACoS can be used for sampling and reconstruction with a variety of degradation operators, such as deblurring, inpainting, and superresolution. First, they show that in these cases, solving the reconstruction problem associated with each degradation is feasible for large-scale image datasets such as CIFAR-10 and CelebA. Then, they show that the blur transformation can be used to sample CelebA and AFHQ images, albeit with significantly reduced sample quality and diversity. They finally show as a proof of concept that other transformations such as inpainting, super-resolution, and animorphosis, can also be used to generate samples.

**Strengths:**

- Clarity: the paper is well written and very clear. To the best of my knowledge the derivations are correct.
- Novelty: the idea behind this paper is interesting and novel to the best of my knowledge. However, as I will discuss in below, the practical value of this idea is unclear.
- Low computational cost: the proposed method is simple and computationally chip, given that it only requires fitting one regression model over the dataset.
- Experimental methodology: the experiments in this paper clearly demonstrate that TACoS can be used for sample generation under a variety of datasets and image transforms. It is interesting that the method can be used to generate samples with arbitrary degradation and reconstruction operators, as opposed to Gaussian noising and denoising via score-matching.

**Weaknesses:**

- Unclear practical value: in my opinion, the practical value of this paper is unclear because the proposed algorithm appears to have low sample quality and diversity. There are many existing methods for deterministic iterative sampling algorithms, notably flow based methods like Continuous Normalizing Flows [1] and Probability Flow ODE [2], which can both eliminate the need for sampling noise and attain high sample quality.
- Unclear applications: it is interesting that the proposed method can use arbitrary image transformations, which may pave the way for other applications beyond sample generation. However, it is currently unclear what these applications may be, which further limits the value of this approach.
- Lack of baselines: the sample generation experiments in Table 4 should also include baseline values, representing the FID that can be attained by existing methods such as vanilla DDPM or GAN based methods.

[1] Building Normalizing Flows with Stochastic Interpolants (Albergo and Vanden-Eijnden, 2023)

[2] Score-Based Generative Modeling through Stochastic Differential Equations (Song et al., 2021)

**Questions:**

What are some potential applications (beyond sampling) for TACoS with non-standard degradation operators like animorphosis?

Note: it's a bit confusing to report RMSE in Tables 1-3 but to then discuss the PSNR in the text. It would be helpful stick to one throughout the paper.

**Limitations:**

The authors have adequately addressed the limitations and potential negative societal impacts of their work.

---

> ### Author Rebuttal · Authors · 2023-08-10
>
> Thank you for your detailed feedback. We address each of your points below.
>
> > Unclear practical value and unclear applications
>
> We agree with the reviewer that cold diffusion does not outcompete the much more highly engineered and compute intensive state-of-the-art Gaussian diffusion models. We also agree that multiple works like [1], [2], [3], [4] remove the need to use random gaussian noise during sampling (i.e. at inference time). Moreover, works like [5] and [6] show that degradations other than gaussian noise are useful as diffusion processes.
>
> This work is not meant to be a SOTA method paper, and we have updated our working draft to clarify this. Instead, this work challenges both theory frameworks where noise is a **critical** component for learning the training distribution and the engineering status-quo that has not been explored beyond Gaussian noise or other stochastic degradations. We believe our experiments effectively demonstrate the surprising fact that entirely noise-free schemes (blur, inpainting, super-resolution) can still work.
>
> This observation is not only of theoretical interest, but it may move the community to explore other kinds of diffusion for common tasks like upsampling, and with further research investments such methods may someday become standard tools. Furthermore, by exploring deterministic degradations beyond Gaussian noise, we open up the possibility of finding the best degradation that will work most effectively for image generation. This represents a significant departure from traditional diffusion models and offers a new pathway for research and experimentation in the field. By expanding the horizons of what's considered in terms of degradation, we may unlock new avenues and insights that can contribute to advancing the state of generative models.
>
>
>
> [1] Building Normalizing Flows with Stochastic Interpolants \
> [2] Score-Based Generative Modeling through Stochastic Differential Equations \
> [3] Denoising Diffusion Implicit Models \
> [4] Elucidating the Design Space of Diffusion-Based Generative Models \
> [5] Structured Denoising Diffusion Models in Discrete State-Spaces \
> [6] DiffusionDet: Diffusion Model for Object Detection
>
>
>
>
>
>
> > Lack of baselines: the sample generation experiments in Table 4 should also include baseline values, representing the FID that can be attained by existing methods such as vanilla DDPM or GAN based methods.
>
> In Table 4 of our paper, we compare our "cold diffusion" which uses a deterministic blur degradation to a noise-based degradation, which we call "hot diffusion", this really is a vanilla diffusion model. The noise-based degradation uses Gaussian noise and the sampling provided in Algorithm 2. Though it may appear different from existing methods like DDPM  or DDIM, as discussed in section 5.1, the sampling method underlying the proposed TACoS is equivalent to DDIM. We present the fact that DDIM is a special case of our Algorithm 2 for the case of Gaussian noise in Appendix A.6. Our revised manuscript now clarifies this.
>
>
>
>
> >What are some potential applications (beyond sampling) for TACoS with non-standard degradation operators like animorphosis?
>
> We present results for various non-standard degradations like animorphosis or snow to show that our proposed sampling algorithm is agnostic to any degradation and is not designed for one specific degradation. Another use of animorphosis can be in building flows between any two arbitrary distributions.
>
> >Note: it's a bit confusing to report RMSE in Tables 1-3 but to then discuss the PSNR in the text. It would be helpful stick to one throughout the paper.
>
> We thank the reviewer for bringing this confusion to our notice. We have now revised our text and use RMSE everywhere, aligning the metrics throughout the paper for consistency.  We'll include these edits in our camera ready version.
>
> Thank you again for your thoughtful review. We made an effort to address your feedback including paper edits and would appreciate it if you would consider raising your score in light of our response.  Do you have any additional questions we can address?

---

> > ### Comment · Reviewer_SFSB · 2023-08-13
> > **Response to rebuttal**
> >
> > Thank you answering my questions and for clarifying the equivalence between hot diffusion and DDIM. I appreciate the authors' efforts to discuss the merits of this paper with me. I feel that my understanding of the contributions and applications has not changed, so I do not plan to edit my score.

---

### Author Rebuttal · Authors · 2023-08-10

We thank all of our reviewers for their thoughtful comments. Based on all the suggestions, we have updated our draft and we would like to highlight a few central contributions of our work.
1.  In this paper, we aim to challenge the common belief that Gaussian noise in any form, either during training or sampling is **necessary** for diffusion models to work. We believe this will not only be of theoretical interest, but it will move the community to investigate other components that could improve diffusion model such as model architecture or training setup.
2. We demonstrate the above point by using different types of **deterministic** degradations. We are enthusiastic readers of other papers that use alternative noise-based degradations like gamma or salt-and-pepper noise, as mentioned, but whether alternative noise-based degradations are also possible is not a question we are investigating in this work.
3. We noticed that where a few questions regarding baselines. We want to highlight that the presented hot diffusion is in fact DDIM. We present the equivalence of our proposed algorithm TACoS in section 5.1 and Appendix A.6. We have clarified this in our revision.

Overall, we believe our findings that cold diffusions can perform high-quality generation is surprising and provides a key insight in the burgeoning study of diffusion models as a whole. Hence, we believe it would be valuable for the community for this work to appear at NeurIPS.

---

### Decision · Program_Chairs · 2023-09-21

**Decision:**

Accept (poster)

**Comment:**

The reviewers and AC appreciated the novelty of the paper and the generality of the proposed deterministic generative diffusion framework. It is clear from the reviewer comments and the discussion that the proposed method is not state-of-the-art in generation in terms of sample quality. The practical uses of the proposed perturbations is not clear as reviewers raised. It is interesting that the proposed method can use arbitrary image transformations, which may open directions for future applications beyond sample generation or solving inverse problems. Animorphosis and snow perturbations are cool examples of what this method can achieve. (A google search suggests also 'Therianthropy' as a suitable term). The paper is very well written and clarifies also related recent work in a very clear framework that will benefit readers.  Therefore,  I think this paper deserves to be accepted for publication in Neurips due to its novelty, clarity and potential for future applications.